# Permeability of membranes in the liquid ordered and liquid disordered phases

An Ghysels [1]*, Andreas Krämer [2], Richard M. Venable[2], Walter E. Teague Jr[3], Edward Lyman[4], Klaus Gawrisch [3] & Richard W. Pastor [2]*

The functional significance of ordered nanodomains (or rafts) in cholesterol rich eukaryotic cell membranes has only begun to be explored. This study exploits the correspondence of cellular rafts and liquid ordered ($L_o$) phases of three-component lipid bilayers to examine permeability. Molecular dynamics simulations of $L_o$ phase dipalmitoylphosphatidylcholine (DPPC), dioleoylphosphatidylcholine (DOPC), and cholesterol show that oxygen and water transit a leaflet through the DOPC and cholesterol rich boundaries of hexagonally packed DPPC microdomains, freely diffuse along the bilayer midplane, and escape the membrane along the boundary regions. Electron paramagnetic resonance experiments provide critical validation: the measured ratio of oxygen concentrations near the midplanes of liquid disordered ($L_d$) and $L_o$ bilayers of DPPC/DOPC/cholesterol is 1.75 ± 0.35, in very good agreement with 1.3 ± 0.3 obtained from simulation. The results show how cellular rafts can be structurally rigid signaling platforms while remaining nearly as permeable to small molecules as the $L_d$ phase.

[1] Center for Molecular Modeling, Ghent University, Technologiepark 46, 9052 Gent, Belgium. [2] Laboratory of Computational Biology, National Heart Lung Blood Institute, National Institutes of Health, Bethesda, MD 20892, USA. [3] Laboratory of Membrane Biochemistry and Biophysics, National Institute on Alcohol Abuse and Alcoholism, National Institutes of Health, Bethesda, MD 20892, USA. [4] Department of Physics and Astronomy and Department of Chemistry and Biochemistry, University of Delaware, Newark 19716 DE, USA. *email: an.ghysels@ugent.be; pastorr@nhlbi.nih.gov

The raft hypothesis posits the existence of ordered micro-domains of lipids and proteins in the plasma membrane that are critical in signaling, protein trafficking, and exo- and endocytosis[1,2]. Cellular rafts are enriched in cholesterol and sphingolipids, and their potential correspondence with the liquid ordered ($L_o$) phase in ternary mixtures of lipids has been recognized since the early 2000s[3,4]. Claims of correspondence were initially considered controversial[5], based mainly on concerns regarding detergent-based biochemical methods for detecting insoluble fractions. Recent experiments[6–9] indicate that the essential structural features of rafts and $L_o$ phases are shared by localized regions in the plasma membrane. In particular, complex mixtures of lipids and proteins extracted directly from live cells phase separate into more and less ordered regions in a manner strikingly similar to simpler ternary mixtures[6], although the ordered regions in such samples are somewhat less ordered than their ternary counterparts[7]. Eggeling et al.[8] detected nanoscale sphingomyelin and cholesterol-dependent heterogeneities in the plasma membranes of living PtK2 cells, and Stone et al.[9] showed co-localization of raft markers with regions of increased order in living immune cells. Hence, it is no longer speculative to assert that a well-designed study on $L_o$ phases may lend insight into the structure and function of cellular rafts.

This report presents molecular dynamics (MD) simulations of permeation of oxygen and water through membranes in the liquid ordered and liquid disordered ($L_d$) phases of dipalmitoyl-phosphatidylcholine (DPPC), dioleoylphosphatidylcholine (DOPC), and cholesterol (chol). Oxygen and water are excellent candidates for such a study. They are small and their permeation can thereby be simulated directly using conventional MD, as opposed to enhanced sampling methods[10] that are not yet validated for liquid ordered phases. Their hydrophobicities span a wide range that includes many permeants of general interest. Last, their permeabilities have been extensively simulated in fluid ($L_\alpha$) phase lipid bilayers, informing the comparison with $L_o$ and $L_d$ phases. In particular, recent simulations have examined the modulation of oxygen permeability by different lipids[11–13] and cholesterol[14–16], and have probed the effects of potential energy functions on water permeability[17,18].

The simulation-based conclusions are supported by electron paramagnetic resonance spectroscopy (EPR). Specifically, the relative concentrations of molecular oxygen in the center of bilayers of $L_o$ and $L_d$ phases were measured by spin-label oximetry[19–22]. Molecular oxygen enhances longitudinal and transverse relaxation rates of stable nitroxide radicals. Samples were doped with a low concentration of the phospholipid 16:0–16 doxyl PC, which introduces stable, doxyl radicals near the center of bilayers.

Of special significance is the simulation result that permeabilities of $O_2$ and water are only factors of three and seven times lower, respectively, in the $L_o$ phase than in the $L_d$ and $L_\alpha$ phases. This is in remarkable contrast to the experimentally observed 24-fold difference for oxygen permeability in dimyristoylphosphatidylcholine (DMPC) gel ($L_\beta$) and $L_\alpha$ phases,[23] and the 2000-fold difference of water permeability in the $L_\beta$ and $L_\alpha$ phases of DPPC.[24] Differences in permeabilities of small solutes in the $L_o$ and $L_\beta$ are to be expected. As shown by the simulations of Sodt et al.[25], the $L_o$ phase of DPPC/DOPC/chol consists of more ordered regions of DPPC surrounded by a layer of cholesterol and less ordered DOPC. The average deuterium order parameters from these simulations agree well with experiment[26], and more recent experimental measurements of single-molecule diffusion support the presence of substructures in the $L_o$ phase[27]. In contrast, the lipids in the DOPC-rich $L_d$ phase are relatively well mixed, and the chain order is comparable with the $L_\alpha$ phase of homogenous bilayers.[26] Simulations have revealed similar

ordered microdomains in $L_o$ phases containing N-palmitoyl sphingomyelin (PSM)[28].

The Results begin with the computation of the oxygen and water permeabilities and related properties for DPPC/DOPC/chol $L_o$ and $L_d$ phases. This is followed by analysis of the permeation pathways, including those of water in $L_o$ and $L_d$ phases containing PSM and 1-palmitoyl-2-oleoylphosphatidylcholine (POPC) from previously published simulations.[28] The results of EPR experiments are then presented and compared with simulation. The Discussion begins with an examination of the model of permeation in the $L_o$ phase that emerges from the simulations. Permeabilities in other phases are compared and related to component surface areas and associated chain properties. The agreement of simulation and related experiments is further analyzed, and some of the implications of the present results to permeation of small solutes in the cell membrane rafts are considered.

## Results

**Simulated permeability and free energy and diffusion.** As detailed further in the Methods section, the simulated and experimental systems presented here are of homogeneous liquid ordered and liquid disordered phases of DPPC/DOPC/cholesterol; i.e., they are not mixtures of the two phases. See Supplementary Table 1 for the compositions. However, while the lipids in the $L_d$ phase are distributed relatively uniformly, the DPPC chains in the $L_o$ phase forms hexagonally packed microdomains.

Figure 1 shows top-down and side views of a single oxygen successfully entering the $L_o$ phase over 500 ps. The highlighted oxygen (in red) enters a region between clusters of hexagonally packed DPPC chains (top panel), and then rapidly diffuses along this interface to the center of the bilayer (middle panel), which contains most of the other oxygens (rendered in purple). The bottom panel of Fig. 1 depicts a coarse-grained representation of the same set of snapshots, where the hydrocarbon chains of DPPC are smoothed and all other molecules except for the permeating oxygen not rendered. As shown in the following subsection, the permeation pathways do indeed proceed along the DOPC/cholesterol boundary regions between the DPPC chains. Supplementary Fig. 3 shows four other ultimately successful entrance trajectories for oxygen. Supplementary Fig. 4 shows four entrances for water into the $L_o$ phase. As for oxygen, water transits typically begin in DOPC/chol regions and can laterally diffuse in the midplane. However, unlike oxygen, which on average remains in the midplane for ~60 ns, water is unstable in the midplane and, on average, escapes the bilayer in 700 ps. A detailed analysis of the oxygen and water characteristic times is presented in the Discussion section.

Figure 2 compares the free energy profiles of oxygen and water for each phase. These are commonly denoted potentials of mean force (PMF) and were calculated from the trajectory as $F(z) = -k_B T \ln p(z)$, where $p(z)$ is the probability of finding the permeant at position $z$, $z$ is the position along the bilayer normal with respect to the membrane center, $T$ is the absolute temperature, and $k_B$ is Boltzmann's constant. These profiles are denoted $F_w(z)$ and $F_{O2}(z)$ for water and oxygen, respectively.

The free energy barrier for oxygen in $L_o$ is located further from the midplane, and is slightly higher than in $L_d$ (consistent with the larger thickness and smaller surface area of $L_o$), and the minimum at the midplane is considerably narrower. The shoulders between $|z| = 10–20$ Å reflect the ordering of the acyl chains. Though the minima at the midplanes are equally deep in both phases, the narrowness of the free energy well in the $L_o$ phase indicates that fewer oxygens are absorbed. The head group region in $L_o$ is less hydrated than in $L_d$, and water is rarely in the

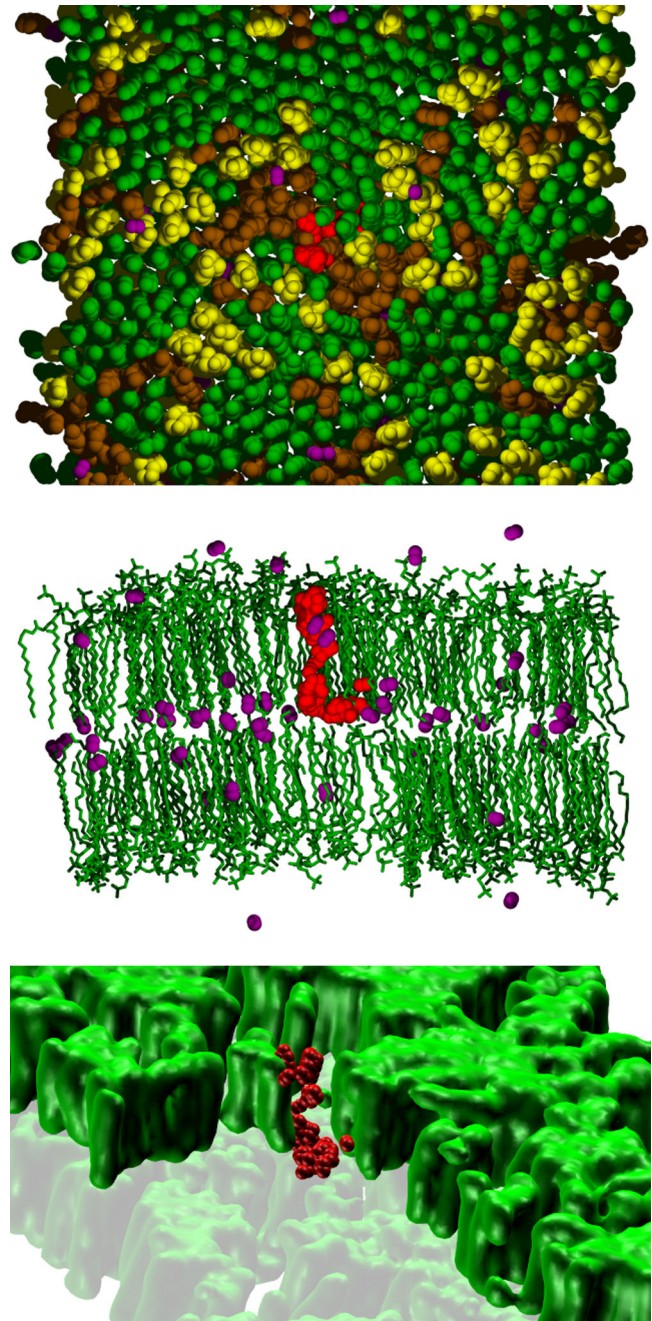

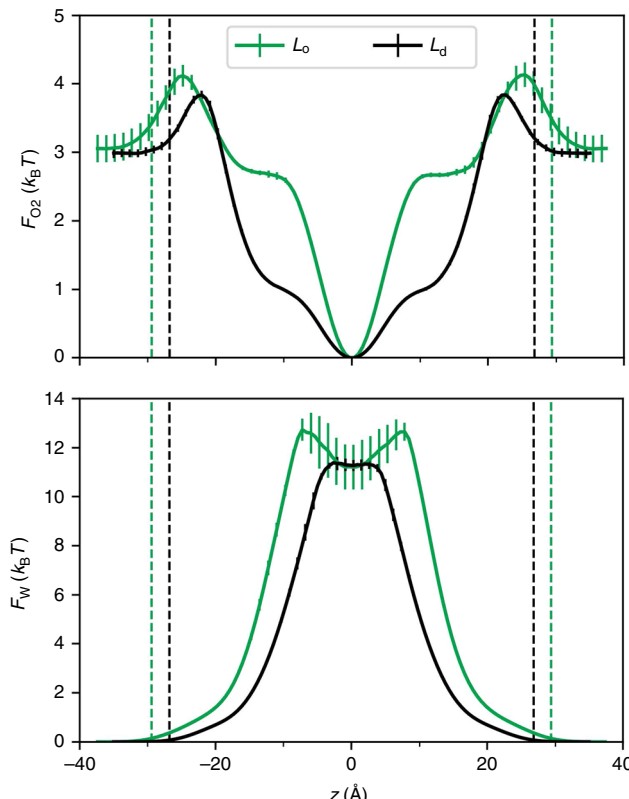

**Fig. 2 Free energy profiles for oxygen and water.** The profiles $F(z)$ were computed from the histogram $p(z)$ for oxygen (top) and water (bottom) as a function of position with respect to the membrane center at $z = 0$. Error bars show the standard deviation over the four replicates. The profiles were not symmetrized, showing the good convergence of the data. The dividing surfaces at $z = \pm h/2$ used in the analyses are denoted with vertical dotted lines. Source data are provided as a Source Data file.

**Fig. 1 Simulation snapshots of a single oxygen entering the liquid ordered phase.** The bilayer is represented by the lipid configuration at the beginning of the time sequence, and waters and ions are removed in all panels. Positions of the diffusing oxygen (in red) are spaced at 5 ps intervals over 500 ps. Top panel: a top-down view with head groups removed to highlight the chain packing (DPPC green, DOPC brown, and cholesterol yellow). Middle panel: side-view with DOPC and cholesterol removed, and with other oxygen molecules from the initial configuration (purple); atoms in only half of the box are included for clarity. Bottom panel: the same set of frames with the single oxygen and DPPC (including the head groups and smoothed) with part of the top leaflet removed and the bottom leaflet in semitransparent to highlight the substructure of the phase.

bilayer for either phase. However, while the barrier for water permeation is precisely at the midplane for $L_d$, there is a metastable minimum in the midplane for $L_o$ which can briefly trap waters (see also Supplementary Fig. 4).

The permeabilities $P$ from Eq. (2) (a counting method) and Eq. (5) (from Bayesian Analysis or BA) are listed in Table 1 (see the Methods section for details). Beginning with oxygen, the number of bilayer crossings for each phase is well over 100 (Supplementary Table 2), allowing estimates with good precision. $P(L_d)/P(L_o)$ (3.5 from counting and 2.9 from BA) indicate a statistically significant difference in permeabilities between the two phases. Differences in the individual values of $P$ between the two methods may arise from assumptions inherent to the BA, which assumes one-dimensional diffusive transport without solvent memory effects. In contrast, the MD trajectories explicitly contain non-diffusive behavior at short timescales and memory effects associated with slow solvent relaxation.[12]

The preceding ratios of permeabilities for $O_2$ in the $L_d$ and $L_o$ phases are approximately twice the value of the ratio of the partition coefficients (Table 1), implying differences in the diffusion profile normal to the membrane surface. The diffusion profiles normal and parallel to the membrane surface are the main output of the BA, and it is profitable to analyze them in detail. Figure 3 (top) shows that $D_\perp(z)$ for the $L_o$ is reduced by ~50% in the midplane, and up to 55% in the chain and head group regions compared with the $L_d$. The $L_o$ is thicker than the $L_d$, which further reduces the permeability.

The BA also provides the diffusion profile parallel to the membrane. As opposed to the permeability through the membrane, the dominant contribution to transport parallel to the membrane comes from the largest values in $D_\parallel(z)$, where simultaneously $F(z)$ is lowest. $D_\parallel(z)$ for $L_o$ in the chain regions is

**Table 1 Permeabilities and partition coefficients from simulation.**

| System (method) | P (cm/s) | | $P(L_d)/P(L_o)$ | $K(L_d)/K(L_o)$ |
|---|---|---|---|---|
| | $L_o$ | $L_d$ | | |
| $O_2$ (counting) | 5.2 (4.6–5.8) | 17.9 (16.3–20.0) | 3.5 (3.0–4.0) | 1.9 (1.6–2.1) |
| $O_2$ (BA) | 9.4 (8.3–10.5) | 27.1 (26.1–28.1) | 2.9 (2.6–3.2) | 1.9 (1.5–2.3) |
| Water (counting) | $0.13 \times 10^{-3}$ (0.06–0.22) | $0.91 \times 10^{-3}$ (0.70–1.1) | 6.8 (3.9–15.3) | 1.339 (1.337–1.341) |
| $h/2$ (Å) | 29.38 | 26.81 | | |

Permeabilities $P$ of oxygen and water in the $L_o$ and $L_d$ phases of DPPC/DOPC/cholesterol, ratios of permeabilities and partition coefficients $K$, and locations of the dividing surfaces with respect to the bilayer center $h/2$. For the BA method, twice an estimate of the standard error is in parentheses. For the counting method, the 95% confidence intervals are listed. Source data are provided as a Source Data file

systematically lower than for $L_d$ (Fig. 3 middle), but the dominant contribution originates from the midplane where $D_\|(z)$ for $L_o$ is slightly larger than that of $L_d$. The cancellation of these effects makes the effective radial permeability $P_\|$ (a measure for radial transport[12]) comparable for $L_o$ and $L_d$, with BA values of 36.5 cm/s and 32.1 cm/s, respectively.

The anisotropy of the diffusion tensor (Fig. 3, bottom) is higher in the $L_o$ phase: the anisotropy ratio $D_\|/D_\perp$ ranges between 4% in the tail region and 300% in the midplane for $L_o$, and between 30 and 170% for the $L_d$. The anisotropy in the head group region is similar for the two phases. The large anisotropy in $L_o$ indicates that the most efficient oxygen paths are normal to the surface in the hydrophobic region, as radial diffusion is considerably slower, whereas oxygen can freely move radially in the midplane (with limited normal diffusion).

A useful measure of radial transport is $L_\|$, which is the distance travelled laterally before the permeant escapes the bilayer[12]. The larger radial diffusion constant in the midplane and more pronounced minimum in $F_{O2}(z)$ (Fig. 2, top) explain the larger value of $L_\|$ for oxygen in the $L_o$ phase (163 Å from the BA and 210 Å directly from the trajectory) than in $L_d$ (149 and 157 Å from the BA and trajectory, respectively). The slower normal diffusion in $L_o$ and complex substructure (described in the following subsection) allow more time for the oxygens to travel laterally than in $L_d$.

The number of water crossings observed in the simulations is considerably lower than that for oxygen, as expected from the high barrier (Fig. 2, bottom; Supplementary Table 2). Nevertheless, the trend is the same: the permeability in the $L_o$ phase is lower than in $L_d$, and the difference is statistically significant. The sevenfold reduction carries a high uncertainty, and nearly includes the threefold reduction found for oxygen. It should be noted that the permeability of water in fluid-phase DOPC simulated with the same methodology[10] is approximately a factor-of-10 lower than the experiment[29], as anticipated for a nonpolarizable force field[10,18]. Consequently, the focus of the following analysis is on the ratios and trends from simulations of different phases rather than on averages from a particular phase.

**Permeation pathways.** The threefold to sevenfold reductions of permeability of oxygen and water listed in Table 1 are statistically significant and far from the experimentally determined ratios for fluid and gel phases noted earlier (24 for oxygen in DMPC[23] and 2000 for water in DPPC[24]). Insight into this difference between $L_o$ and gel phases can be obtained by examining where these permeants reside in the $L_o$ and $L_d$ phases along their translocation pathways.

To this end, the locations of oxygen and water in the two phases were analyzed by determining their three nearest

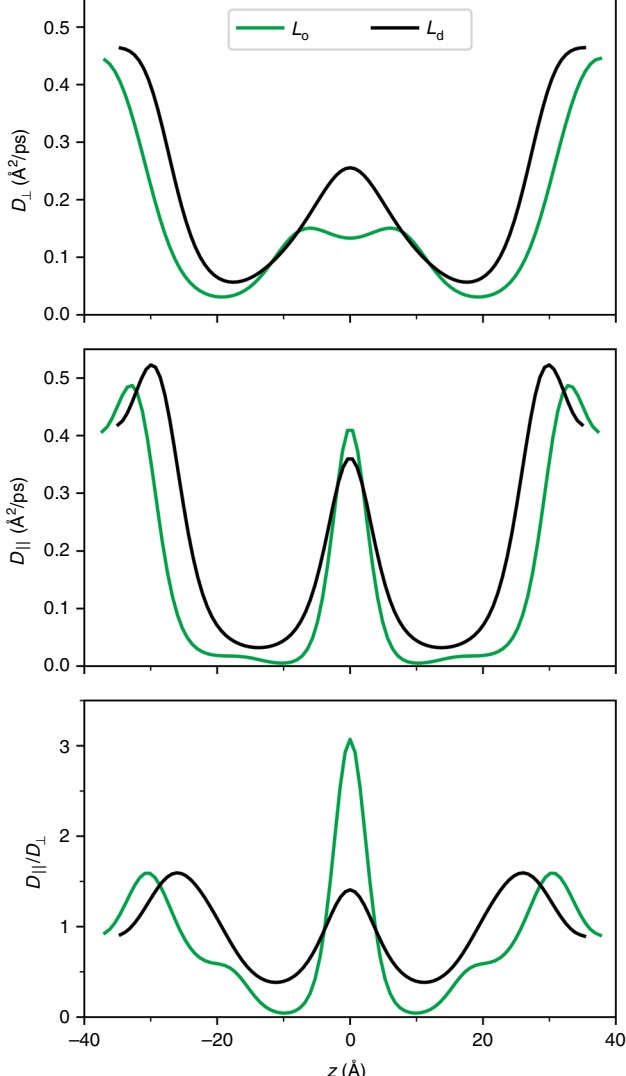

**Fig. 3 Diffusion profiles for oxygen.** Profiles normal $D_\perp(z)$ (top) and parallel $D_\|(z)$ (center) to the membrane surface in the $L_o$ and $L_d$ phases were obtained from BA. The anisotropy is the ratio $D_\|(z)/D_\perp(z)$ (bottom). Source data are provided as a Source Data file.

neighbors along the bilayer normal. This begins by calculating the three closest lipid chains to the permeant, where the distance is defined as the minimum distance between the permeant's heavy atoms and the chain carbons. The three nearest neighbors were sorted into four classes: all lipids of the same kind (denoted 3-DPPC, 3-DOPC, and 3-chol), and all other combinations of DPPC, DOPC, and chol, denoted Mix. Hence, the boundary region of the DPPC microdomains includes 3-DOPC, 3-chol, and Mix. Values were determined for each 1 Å slab, and normalized; i.e., the probabilities of 3-DPPC, 3-DOPC, 3-chol, and Mix sum to 1.0.

The results of the three nearest-neighbor analysis are plotted in Fig. 4. The profiles for oxygen and water in the $L_o$ phase are remarkably similar, given the increased statistical error in the water counts (this is evident in the asymmetry of the profiles). The fraction of 3-DPPC is ~0.45 at $|z| = 25$ Å (the approximate location of the phosphate peak), indicating no strong lipid preference at the bilayer surface. However, 3-DPPC decreases in the chain regions (20 Å > $|z|$ > 5 Å), and Mix becomes the dominant population. Both fractions revert to the head group

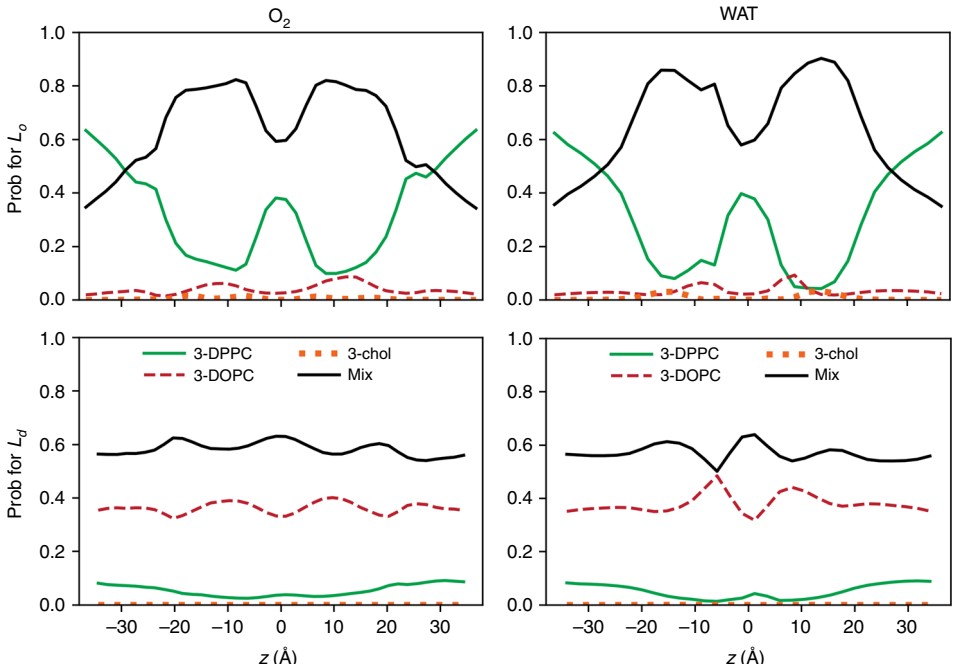

**Fig. 4 Three nearest-neighbor analysis for permeation in DPPC/DOPC/chol.** The probabilities of the classes of three nearest neighbors are shown for oxygen (left) and water (right) in the $L_o$ and $L_d$ phases. See the text for definitions of the classes. Source data are provided as a Source Data file.

levels in the bilayer midplane ($|z| < 5$ Å). Fractions of 3-DOPC and 3-chol are low throughout the bilayer as expected from the relatively low populations of each of these lipids, though 3-DOPC increases to ~10% at $|z| \approx 12$ Å, near the minimum of 3-DPPC. In summary, ~90% of the permeation paths avoid the DPPC hexagonally packed chain regions, freely sample the midplane, and escape in the same manner as entering. The fraction profiles for the $L_d$ phase are nearly constant (bottom row of Fig. 4), indicating no particular preference for a given lipid type.

The water analysis was also performed on previously published[28] simulations carried out on the Anton supercomputer[30] with no added oxygen (see Supplementary Table 3 for compositions and other system details). PSM takes the role of the DPPC in four of the six systems, and the substitution of POPC for DOPC was investigated. The relative lipid concentrations in these ternary systems yield homogeneous $L_o$ or $L_d$ phases. The free energy profiles are very similar to the two DPPC/DOPC/chol systems already described (Fig. 5, top). In particular, $F_w(z)$ for DPPC/DOPC/chol with and without oxygens are virtually identical, indicating that oxygen is not perturbing the structures. The three nearest-neighbor analysis (Fig. 5, middle, bottom) also shows the same behavior: the ordered subdomains of DPPC and PSM exclude water in their chain regions, but allow it in the midplane.

**EPR experimental**. The EPR spectra of 16:0–16 Doxyl PC in the $L_d$ and $L_o$ phases (Fig. 6) are indicative of homogeneous populations of the 16:0–16 Doxyl PC label, suggesting that the probes are in single phases. The spectra indicate a high level of motional freedom of the label as expected for a location near the center of the bilayer. The additional broadening of the $M_I = \pm 1$ resonances in the $L_o$ phase compared with $L_d$ is a reflection of an increase in motional correlation times and/or increased anisotropy of label motions in the ordered phase.

In the presence of molecular oxygen, the $R_1$ and $R_2$ relaxation rates increase due to Heisenberg exchange broadening of the resonance of the stable nitroxide radical[19]. The increase $R_2$ was

measured directly from broadening of the $M_I = 0$ resonance at low applied microwave power (1.6 mW) to avoid line broadening from saturation effects. The increase of linewidth between samples exposed to $N_2$ and pure $O_2$ is reported. The ratio of oxygen-induced transverse relaxation enhancement $R_{2,Ld}/R_{2,Lo} = 1.6 \pm 0.5$.

An alternative approach is to measure the signal amplitudes of the $M_I = 0$ resonance in a power-saturation experiment[21,22]. The power-saturation parameter $P_{1/2}$ is defined as the microwave power, where the first derivative amplitude is reduced to half of its theoretical unsaturated value determined by extrapolating the linear region[22] (Fig. 7). It was shown that the parameter $P_{1/2} \propto R_1 R_2$. Dividing $P_{1/2}$ by the linewidth, which is proportional to $R_2$, provides a parameter solely proportional to $R_1$ irrespective of variations in nitroxide mobility (lineshape). The longitudinal relaxation rate $R_1$ is proportional to the Heisenberg exchange frequency from collisions of the spin label with oxygen molecules[22]. Power saturation was measured by comparing samples exposed to $N_2$ and air (21% $O_2$) (Fig. 7). The reason for measuring in air (rather than pure oxygen) was to limit relaxation enhancement. Samples exposed to pure oxygen had insufficient power saturation for accurate measurements at up to 50 mW microwave power. Higher power levels resulted in sample heating and related instrument drift. The oxygen-induced longitudinal relaxation enhancement $R_{1,Ld}/R_{1,Lo} = 1.9 \pm 0.5$ (see Supplementary Information for more details).

**Comparison of simulations and EPR experiments**. The enhancement of longitudinal and transverse relaxation rates of the nitroxide spin probe by oxygen is quantum mechanical in nature[19] and therefore cannot be directly simulated with classical MD. Furthermore, spin probes were not included in the simulation. An ameliorating consideration is that only the ratio of the relaxation rates in the $L_d$ and $L_o$ phases is required for this study, so the simplifying assumptions described below can be anticipated to apply equally to both phases and therefore cancel.

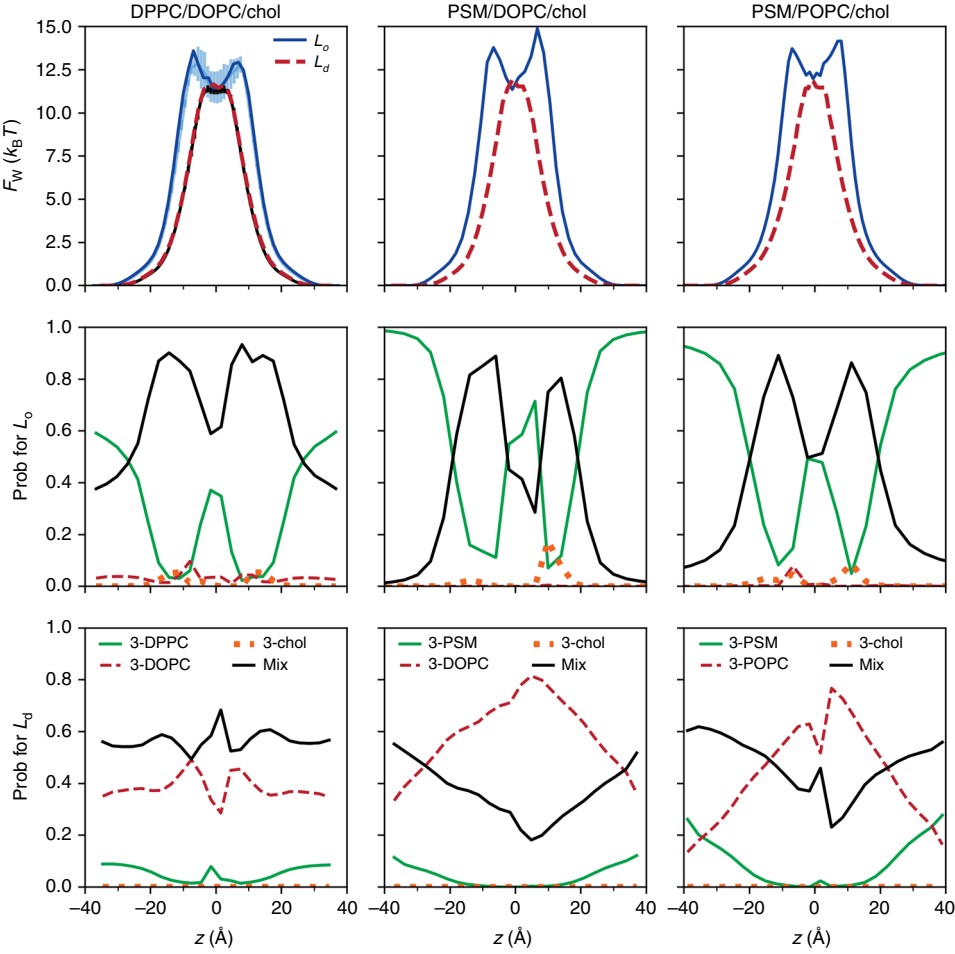

**Fig. 5 Comparison of permeation through PSM-containing membranes.** Free energy profiles for water permeant (top row), probabilities from the three-neighbor analysis of the liquid ordered (middle row) and liquid disordered (bottom row) phases from Anton simulations of DPPC/DOPC/chol (left column), PSM/DOPC/chol (middle column), and PSM/POPC/chol (right column). The light blue and black lines in the top left panel are the PMFs with error bars from the CHARMM simulation of DPPC/DOPC/chol which also included oxygen (Fig. 2). The statistical errors are higher than in Fig. 4 because of the smaller number of saved coordinate sets. Source data are provided as a Source Data file.

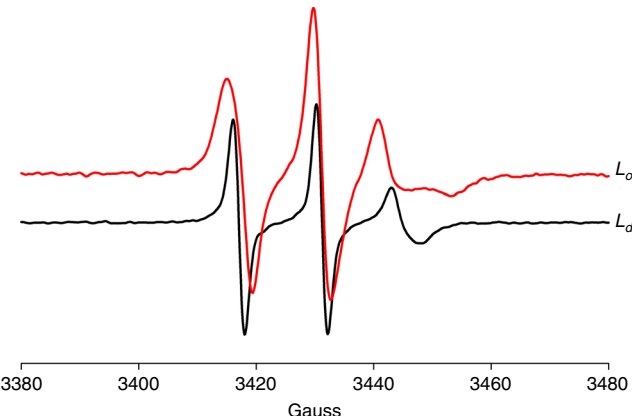

**Fig. 6 EPR spectra of DPPC/DOPC/chol.** Spectra are of 16:0–16 Doxyl PC, added to DPPC at 1 mol%, in DPPC/DOPC/chol, 0.24/0.69/0.07 ($L_o$, red), and DPPC/DOPC/chol, 0.24/0.69/0.07 ($L_d$, black), recorded at 298 K in pure nitrogen. The increased linewidth, in particular of the $M_I = -1$ (right) resonance in the spectrum, is a reflection of a higher immobilization of the label in the $L_o$ phase. Source data are provided as a Source Data file.

As a first assumption, the methyl group of chain 2 of DPPC is used as the surrogate for the 16:0–16 Doxyl PC spin probe. While the N–O radical (the site of the Heisenberg exchange interaction) is positioned near the methyl group, some differences between chains with and without the probe are likely.

As a second assumption, the magnitude of the interaction is estimated from the population of oxygen near the methyl group, in practice evaluated as the oxygen population within a contact distance $r$ of the methyl carbon atom of chain 2 of DPPC; this quantity is designated $n_{O2}(r)$. This assumption is consistent with theory which predicts that collisions at the 4–5 Å length range are responsible for quenching[19,31], but clearly some details of oxygen and probe interactions are missing. See the Methods section for more details on the explicit evaluation of $n_{O2}(r)$ from the simulation.

Last, $n_{O2}(r)$ is normalized by the oxygen concentration in water, $c_w$ (Eq. (4) and associated discussion in the Methods section). As opposed to experiments, $c_w$ is not the same in the simulated $L_o$ and $L_d$ phases (the number of oxygens in each simulation system is the same, but the partition coefficients are not).

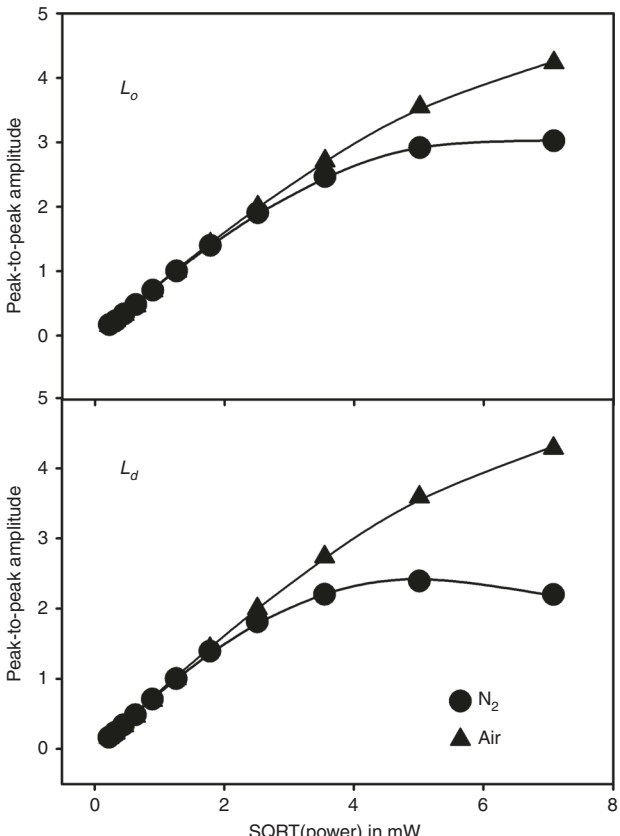

**Fig. 7 Power-saturation curves of 16:0–16 Doxyl PC in $L_o$ and $L_d$ phases.** Membranes were exposed to $N_2$ (circles) and air (21% $O_2$, triangles). The peak-to-peak amplitude of the $M_I = 0$ resonance (center peak in Fig. 6) was measured as a function of microwave power from 50 to 0.05 mW. The line is a fit to the experimental data as described by Eq. (8). Source data are provided as a Source Data file.

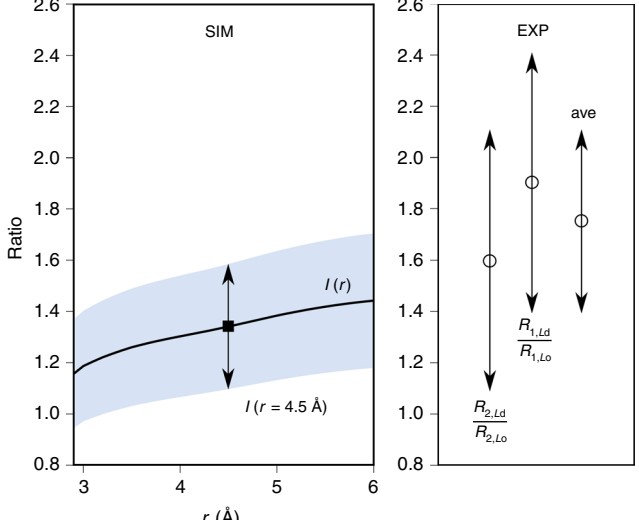

**Fig. 8 Relative oxygen population near bilayer midplanes.** Left panel: simulated $I(r)$, the ratio of (normalized) oxygen population near the probe in the $L_d$ and $L_o$ phases (Eq. (1)) as a function of the interaction radius $r$ between the methyl carbon of chain 2 of DPPC and oxygen, with $I(r = 4.5$ Å) shown explicitly (filled square). The blue band is twice the standard error over the four replicas. Right panel: experimental ratios of $R_{1,Ld}/R_{1,Lo}$, $R_{2,Ld}/R_{2,Lo}$, and their average (open circles). The uncertainties are standard errors from the fits. The source data are provided as a Source Data file.

The simulation surrogate of the experimental relaxation ratios $R_{Ld}/R_{Lo}$ can now be calculated as

$$I(r) = \left(\frac{n_{O2}(r)}{c_w}\right)_{L_d} / \left(\frac{n_{O2}(r)}{c_w}\right)_{L_o}. \quad (1)$$

The left panel of Fig. 8 indicates that $I(r)$ is relatively constant in the 4–5 Å range, and highlights the value at 4.5 Å[31]: $1.34 \pm 0.24$, where the uncertainty is twice the standard error over four replicates. The right panel of Fig. 8 plots $R_{1,Ld}/R_{1,Lo}$, $R_{2,Ld}/R_{2,Lo}$, and their average ($1.75 \pm 0.35$). The agreement of simulation and experiment is very good, considering the uncertainties of each method and assumptions associated with the estimate from simulation.

## Discussion

The structural differences of the $L_o$ and $L_d$ phases suggest two simple models for the origin of their permeability differences. The first, M1, is that all regions of the $L_o$ are permeable, but somewhat reduced compared with the thinner $L_d$ phase. The second, M2, is that permeants are completely blocked in the DPPC regions and only transit along the boundary regions between the DPPC clusters. Behavior supporting M2 has been observed in simulations of biphasic bilayers, where enhanced permeability of water, ions, and larger solutes occurs at the interface between the gel and the fluid phases[32–34]. However, the interleaflet space between DPPC- or PSM-rich regions in the $L_o$ phase might be accessible to permeants even when the chain regions are not. For example, Del

Regno and Notman[35] proposed that relatively unencumbered lateral diffusion of water in the interleaflet spaces to cholesterol-rich regions is a critical feature of permeability in the stratum corneum.

The simulations reported here indicate that both simple models are incorrect. Oxygen and water are clearly differentially excluded from the DPPC-rich chain regions of the $L_o$ phase, a contradiction of M1, and freely sample all the lipid regions in the midplane before escaping the bilayer, a contradiction of M2 (Fig. 4). The same conclusions hold for water permeation in PSM-containing $L_o$ phases (Fig. 5). The model that emerges from the simulations is that small permeants probe the surface of the $L_o$ phase membrane in a relatively uniform manner, translocate the leaflets along the cholesterol and unsaturated chains and avoid the hexagonally packed saturated acyl chains. Because the space between the leaflets contains substantial free volume, a translocating permeant freely diffuses along the midplane until it leaves the membrane via another break in the lattice. A liquid ordered membrane leaflet might then be imaged as an array of irregularly shaped islands (the hexagonally packed saturated chains) separated by channels (the cholesterol and unsaturated chains). This is shown most clearly in the bottom panel of Fig. 1, where the DOPC and cholesterol are not rendered for clarity and the trajectory of a single oxygen is followed for 500 ps. Note that the two leaflets need not be precisely aligned. This does not substantially change the trajectories of oxygen, which have long residence times in the midplane. However, the path of water through the $L_o$ phase typically involves a short, but noticeable detour in the midplane resulting from blockage by the DPPC-rich regions (Supplementary Fig. 4); this is reflected by the metastable minimum in $F_w(z)$ at the midplane (Fig. 2, bottom).

In contrast to the $L_o$ phase, the behavior of both oxygen and water in the liquid disordered phase is similar to single-component fluid-phase bilayers at the same temperature. For example, the permeability of oxygen from the BA for the $L_d$

(27.1 ± 1 cm/s) brackets 27.5 cm/s, recently calculated for a DOPC bilayer at the same temperature using the same methodology[12]. The free energy and diffusion profiles are also similar to those obtained in recent simulations by other groups using different simulation programs and potential energy functions[14–16], providing additional confidence in the comparison. Likewise, the permeability of water from counting for the $L_d$ ($0.9 \pm 0.2 \times 10^{-3}$ cm/s) is only slightly lower than the $1.5 \pm 0.4 \times 10^{-3}$ cm/s for pure DOPC[10]. The three-nearest-neighbor analysis (Fig. 4, bottom) indicates little preference for a given lipid type, consistent with the fact that the permeant does not need to avoid dense microdomains.

Permeability of water in $L_\alpha$ phases of pure lipids is known to correlate with membrane surface area[36] and it is fruitful to consider area and related properties for the systems studied here. Supplementary Table 4 lists the component surface areas per lipid $A_\ell$ for each lipid in each phase. Values for the multicomponent $L_o$ and $L_d$ were evaluated from Voronoi tessellation[37], and those for the $L_\alpha$ phase of pure lipids directly from previously published trajectories[38]. $A_\ell$ for DPPC and DOPC in their liquid disordered phase and in their homogenous fluid phases are comparable, as is the chain order as measured by the fraction of gauche conformers and deuterium order parameters $S_{CD}$ (Supplementary Figs. 6, 7). (The $L_\alpha$ phase simulations of DPPC were carried out at 323 K, and are thereby expected to be more disordered than DPPC in the $L_d$ phase at 298 K.)

The liquid ordered phase is more condensed. In particular, $A_\ell$ for DPPC in $L_o$ is only slightly higher than the 47.3 Å² determined experimentally for the gel phase at 298 K[39]. The low surface area is reflected by the low population of gauche dihedral angles in the acyl chains (Supplementary Fig. 6), as well as the high deuterium order parameters of carbons 4–10 ($S_{CD} \approx 0.42$; Supplementary Fig. 7). These surface areas, gauche populations, and order parameters are all consistent with tight packing and low free volume in the chain region of $L_o$. However, while the $L_o$ DPPC chains are nearly all trans, they do not have the 32° systematic tilt[39], and interdigitation between leaflets[40] deduced for DPPC in the gel phase. Rather, the chains are relatively well aligned along the bilayer normal in $L_o$ with an average tilt angle of 10° (Supplementary Fig. 8), and the bilayer leaflets are well separated (middle panel of Fig. 1; Supplementary Figs. 3, 4).

The DOPC chains in the $L_o$ phase are less ordered than those of DPPC in this phase. The fraction of gauche conformers in DOPC chains is higher (Supplementary Fig. 6), and the order parameters of carbons 4–7 are lower (Supplementary Fig. 7; the cis double bond at positions 9 and 10 of the DOPC chains makes further comparison difficult). Hence, the average deuterium order parameter in $L_o$, 0.36 from the experiment[26] and 0.37 from simulation[25], is the weighted average from densely packed DPPC and less dense DOPC. The hydroxyl group of cholesterol attracts water close to the chain regions (Fig. 9, top), and may enhance water permeation as well.

It is useful to consider two characteristic times of a permeant: the entrance time $\tau_{entr}$ is the average duration of a successful trajectory leaving the membrane surface and reaching the midplane, and the escape time $\tau_{esc}$ is the average time between entering the midplane region and escaping to the surface. These are listed in Supplementary Table 5 for oxygen and water for the present simulations of $L_o$ and $L_d$ phases, and previous simulations of DOPC in the $L_\alpha$ phase (homogeneous DPPC is not useful here because it is a gel at 298 K). Beginning with oxygen, $\tau_{entr}$ in $L_o$ is twice that of $L_d$, consistent with the lower diffusion constant (Fig. 3) and tighter chain packing. As expected from the free energy minimum in the midplane (Fig. 2, top), $\tau_{esc} \gg \tau_{entr}$ (oxygen is trapped). The characteristic times listed in Supplementary Table 5 were computed directly from the trajectory.

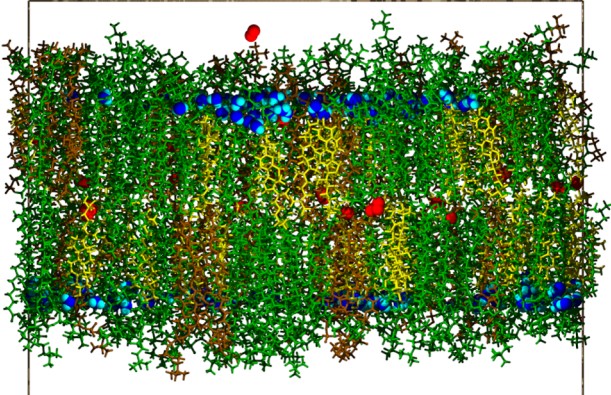

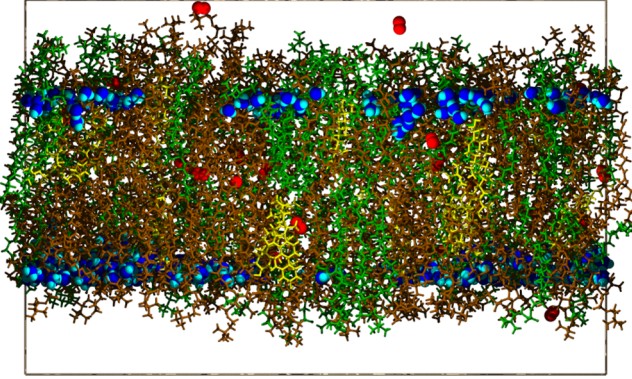

**Fig. 9 Simulation snapshots of DPPC/DOPC/chol.** The liquid ordered and liquid disordered are shown in the top and bottom, respectively. Coloring as follows: DPPC (green); DOPC (brown); cholesterol (yellow); oxygen (red); waters between the phosphate planes of the bilayer (blue oxygen and cyan hydrogen). Other waters, all ions, and atoms in half of the box are omitted for clarity. The boundaries of the periodic cells are shown in solid lines.

Those from the BA followed the same trends, but were slightly lower than found for homogeneous phases[12]. The results for water follow the same trend, although the characteristic escape times are much shorter because the free energy for water is highly unfavorable in the bilayer interior. Note the effect of the metastable minimum in the $L_o$ midplane for water (Fig. 2 bottom), where the difference $\tau_{esc} - \tau_{entr} = 0.56$ ns. This partly arises from the misalignment of entrance and escape pathways most evident in the lower panel of Fig. 1; lateral diffusion in the midplane for water can be seen in some of the trajectories in Supplementary Fig. 4. $\tau_{entr} \approx \tau_{esc}$ for the $L_\alpha$ phase of DOPC, which is consistent with the lack of trapping in the midplane.

The ratio of permeabilities of $L_d$ and $L_o$ phases obtained from the simulations is ~3 for oxygen and 7 for water (Table 1). Hence, even though the hexagonally packed DPPC chains block a substantial fraction of translocation pathways, the net effect on permeation is relatively small. This is because diffusion is not well blocked by partial barriers. A similar effect was shown to be critical for understanding fencing of $PIP_2$ on cell surfaces[41,42].

The very good agreement shown in Fig. 8 between EPR experiments for the ratios of oxygen-triggered Heisenberg exchange in the $L_d$ and $L_o$ phases, and the oxygen populations obtained from simulation is an essential result of this paper. It is important to note that the EPR experiments did not yield absolute concentrations of $O_2$, and that the concentrations of $O_2$ in the simulations are larger than those in the experiment (first subsection of Supplementary Methods). However, the relative insensitivity of $I(r)$ to the contact radius (Fig. 8) allays the

preceding concerns. The critical observation is that the relevant ratios are in the range of 1–2 both in experiment and simulation, indicating that oxygen accumulates in the center of both phases at significant concentrations.

The simulations also provide insight into the probe location in the liquid ordered phase, assuming that the terminal methyl group of the chain is an adequate surrogate of the probe. Specifically, the free energy well for oxygen is almost equally deep at $z = 0$ (Fig. 2, top), meaning that the local concentration is approximately the same at the very center of the $L_d$ and $L_o$ phases. Nevertheless, the ratio is statistically different from 1 in both experiment and simulations. This originates from the flexibility of the probe location, and that it is not located exactly at the center of the membrane. Instead, the average positions of the C16 atom of the C2 chains in DPPC calculated directly from the MD trajectories are displaced from the midplane, with $z = 3.0$ Å in $L_o$ and $z = 2.4$ Å in $L_d$, indicating that they sample different parts of the free energy profiles. Furthermore, the oxygen free energy profiles differ substantially 2–4 Å from the midplane. Hence, it is not surprising that the concentration ratios are statistically different from 1 in simulations.

It is now useful to consider the partition coefficients and permeabilities for gel phase and two component bilayers that show a degree of ordered behavior. Recent experimental measurements[43] of partition coefficients for oxygen in DPPC at 298 K (gel) and 315 K (fluid phase, just above the main phase transition) yield a fluid to gel ratio $K(L_\alpha)/K(L_\gamma)$ of ~10. The preceding ratio is not surprising given the higher density of the gel compared with the fluid phase. However, it is also substantially larger than $K(L_d)/K(L_o) = 1.9$ obtained from the present simulations of DPPC/DOPC/chol at 298 K (Table 1). While tight DPPC chain packing reduces partitioning of oxygen in both $L_o$ (present results) and gel[44] phases, the leaflets in the $L_o$ phase are not interdigitated and contain regions of less structured DOPC, both of which increase free volume (and thereby partitioning) in the $L_o$ compared with the gel. Cholesterol uniformly condenses bilayers composed of saturated lipids. Those bilayers with ~25% cholesterol and higher are commonly denoted liquid ordered[45], and oxygen permeability is substantially reduced[23,46]. The ternary $L_o$ membranes studied here contain localized pathways enriched in a low melting lipid with lower order, and such pathways are absent in binary systems of saturated lipids and cholesterol. Hence, experimental values for the permeability and partition coefficients of oxygen, water, and other solutes in the liquid ordered phases of ternary mixtures will be most useful. Two methodological issues regarding the interpretation of experiment bear noting. First, although $K$ itself is ambiguous because of its bilayer thickness dependence[47,48], the ratio of the partition coefficients of the $L_o$ and $L_d$ phases is relatively insensitive to assumptions regarding the membrane thickness (dashed line in Supplementary Fig. 2). Second, it is possible that some experimental permeabilities contain artifacts from probe molecules, which can disrupt the lattice and provide defects that enhance permeation.

The combination of low free energy pathways for small permeants from the surface to the midplane and nearly free lateral translation in the midplane has important implications for the function of lipid rafts. Rafts would not be expected to seal bilayers to permeants as observed for a gel phase. However, rafts do effectively confine proteins, and the present model is relevant in this regard. For example, Hansen et al. have proposed that the first step in the action of general anesthesia involves the disruption of a raft domain and release of phospholipase D2[49]; this is followed by activation of the potassium channel TREK-1[50]. The structure and dynamics of the $L_o$ phase elucidated here suggest that rafts are quite porous, and that a small anesthetic would be able to rapidly diffuse into the more ordered DPPC or PSM

microdomains along the midplane and more effectively disrupt the phase. It is clear, of course, that a wide range of other permeants needs to be examined to assess the generality of the observations presented here. Furthermore, proteins present in cellular rafts but not included in the present simulations would be expected to modulate the permeation pathways for certain anesthetics.

In summary, the rules of diffusion in liquid ordered phases and biological rafts differ from those in fluid-phase and gel-phase membranes, and elucidating them is a challenge well suited to computer simulation.

## Methods

**Molecular dynamics simulations.** In total, 50 oxygen molecules were added to previously simulated[25] $L_o$ and $L_d$ DPPC/DOPC/cholesterol mixtures (see Supplementary Table 1 for compositions and other relevant system details). NPT simulations of 100 ns were then carried out in CHARMM[51] with the C36 lipid force field[52] to determine the average cell dimensions following the protocol detailed in Ghysels et al.[11] Four trajectories for $L_o$ (285 ns each) and four for $L_d$ (200–240 ns) were generated at constant volume and a temperature of 298 K; 20 ns from one of the $L_o$ trajectories were not used for analysis of oxygen diffusion because of equilibration concerns, but were deemed to be acceptable for water diffusion. The snapshots in Fig. 9 show the packing differences of the $L_o$ and $L_d$ phases, and the water content in their head group regions.

Six systems (Supplementary Table 3) previously generated on the Anton supercomputer[30] were also analyzed. These include DPPC/DOPC/chol[25] for comparison with the simulations containing oxygen presented here and PSM with two different unsaturated lipids[28] to examine effects of lipid type. Because of storage limitations, coordinates for these trajectories were only saved at 250 ps intervals. Hence, accurate computations of water permeability are not possible because a significant number of waters transit the bilayer in <250 ps[10] and are therefore missed. Nevertheless, the equilibrium populations of water in the bilayer were sampled sufficiently to estimate potentials of mean force and examine the translocation pathways.

The number of $O_2$ for these simulations, 50 in each system, was chosen to provide good sampling of transitions in the available computer time rather than an accurate correspondence to the experimental concentration. Previous simulations[11] of water/hexadecane systems with oxygen concentrations comparable with the present system revealed negligible interactions among the $O_2$ molecules, supporting the concentrations used here. A rough estimate of the numbers at the experimental condition of 100% oxygen at 1 atmosphere pressure and 295 K yields ~1.5 $O_2$ molecules for the $L_o$ and $L_d$ system sizes listed in Table 1 (see Supplementary Methods for details). While assumptions regarding the relative solubilities of the two phases and errors in the force field may change the estimate by a factor of 2–3, 1.5 $O_2$ molecules is far less than the 50 in the simulated systems. Hence, the ratios of assorted quantities related to concentration are the focus of this study.

**Evaluation of permeability.** Permeabilities $P$ for both water and oxygen were evaluated using a counting method[10,32,53,54]:

$$P = \frac{r}{2c_w}, \qquad (2)$$

where $r$ is the crossing rate of the permeants (the number of crossings through the membrane in either direction per unit of time and area), and $c_w$ is the concentration of permeant in the water phase. The rate was evaluated as the number of crossings in both directions divided by the length of the simulation $T_{sim}$ and the cross-sectional area $A$ of the bilayer (the surface area of one leaflet):

$$r = \frac{\text{number of crossings}}{A T_{sim}}. \qquad (3)$$

Water crossings were evaluated following the procedure introduced in Ref. [10]: a water transit is tabulated when it enters the membrane, crosses the central region of $|z| < 4$ Å, and exits from the other side of the membrane. The two dividing surfaces for "entering" or "exiting" the membrane are located at $|z| = h/2$, where $h$ is reported in Table 1.

The evaluation of oxygen crossings requires a modification of the procedure above. Indeed, most of the permeants are already in the membrane after equilibration and consequently, the first exit of such an oxygen is not counted as a transition. This leads to an underestimate of the rate when using Eq. (3). A practical solution to this problem for oxygen molecules arises from the two assumptions: (1) permeants inside the bilayer at the start of the simulation have entered the bilayer from one of two sides, but the particular side is not known; (2) permeants lose memory inside the membrane and randomly exit on either the same side or the other side. It follows that the initial escape of such a permeant out of the membrane will increase the total number of crossings by 0.5, on average. The counting of oxygen crossings was modified to let those initial escapes count for only a 0.5 crossing instead of 1 crossing. This modification is not necessary for

water, because the water population inside the membrane at the start of each simulation is negligible.

The concentration of permeant in the water phase $c_w$ is

$$c_w = \frac{N}{A} e^{-\beta F_{ref}} \left( \int_{-\frac{H}{2}}^{\frac{H}{2}} e^{-\beta F(z)} dz \right)^{-1} \qquad (4)$$

where $N$ is the number of permeants, $H$ is the height of the simulation cell, $\beta = 1/(k_B T)$, and $F_{ref}$ is the (constant) value of the free energy profile in the water phase (Krämer et al., in preparation). The $c_w$ may also be obtained by averaging the number of permeants over time in a volume sufficiently far from the membrane where the concentration of the permeant is constant. The values of $c_w$ for oxygen and water are reported in Supplementary Table 7.

Equation 2 is based on Fick's Law (which calculates the permeability from a net flux in one direction) and is nearly assumption-free. The analysis does not require, or provide, information on the diffusion profile of the solute in the membrane.

Permeabilities for oxygen were also evaluated from the inhomogeneous solubility diffusion equation[55]:

$$\frac{1}{P} = e^{-\beta F_{ref}} \int_{-h/2}^{+h/2} \frac{1}{e^{-\beta F(z)} D_\perp(z)} dz. \qquad (5)$$

Here, $D_\perp(z)$ and $F(z)$ were obtained from a BA[11]. This BA also provides $D_\parallel(z)$. As a consistency check, $F(z)$ was also calculated directly from the trajectory from $p(z)$. The diffusion and free energy profiles yield assorted characteristic relaxation times and distances[13], including $L_\parallel$, the lateral distance travelled before escaping.

Equation 5 assumes that the permeation is diffusive and one-dimensional, and is therefore not expected to yield the results identical to Eq. (2). Nevertheless, the diffusion profiles provided by the BA are a valuable addition to the analysis of oxygen permeation (there was insufficient sampling to apply the method to water).

Settings used were the same as for previous studies[11,12]: 100 $z$-bins, 10 cosine basis set functions for the free energy and 6 for the diffusion profiles, 50 Bessel functions for the radial diffusion profile using 100 radial bins and a radial cutoff of 50 Å, and the fit to infinite lag time was based on lag times of 20, 30, 40, and 50 ps. The free energy profiles from the histogram and the BA were essentially identical. Standard errors on BA results were estimated from the standard deviation among the four replicates. This is conservative given the nonlinear nature of Bayesian inference[56].

**Evaluation of partition coefficients**. The partition coefficient $K$ is defined as the ratio of the concentration of the permeant in the membrane ($c_m$) and in water ($c_w$). While $c_w$ (Eq. (4)) is independent of membrane thickness $h$, $c_m$ is not:

$$c_m(h) = \frac{N}{Ah} \int_{-\frac{h}{2}}^{\frac{h}{2}} e^{-\beta F(z)} dz \left( \int_{-\frac{H}{2}}^{\frac{H}{2}} e^{-\beta F(z)} dz \right)^{-1}, \qquad (6)$$

where $Ah = V_m$ is the volume of the membrane. The partition coefficient is then

$$K(h) = \frac{c_m(h)}{c_w} = \frac{1}{h} \int_{-\frac{h}{2}}^{\frac{h}{2}} e^{-\beta(F(z)-F_{ref})} dz. \qquad (7)$$

Hence, an assumption for the value of $h$ is required to estimate the partition coefficient. Here $h/2$ is set to the dividing surfaces used for the evaluation of transitions (see Table 1). While the individual partition coefficients vary as a function of $h$ (Supplementary Fig. 2, axis on left for oxygen), the ratio $K(L_d)/K(L_o)$ is nearly independent (Supplementary Fig. 2, axis on right). The values of $K$ listed in Supplementary Table 7 and ratios in Table 1 are obtained with $h = 29.38$ Å and 26.81 Å for the $L_o$ and $L_d$ phases, respectively.

**EPR sample preparation and measurements**. DPPC, DOPC, cholesterol, and 1-palmitoyl-2-stearoyl-(16-doxyl)-sn-glycero-3-phosphocholine (16:0–16 Doxyl PC) were purchased from Avanti Polar Lipids (Alabaster, AL). Lipids were mixed in stabilized chloroform at a molar ratio of DPPC/DOPC/chol, 0.58/0.09/0.33 (liquid ordered state), and DPPC/DOPC/chol, 0.24/0.69/0.07 (liquid disordered state). The preceding compositions are based on recent neutron scattering results (Dorell et al., in preparation) and assure that each phase is homogenous; i.e., there is no $L_d$ in the DPPC-rich phase and no $L_o$ in the DOPC-rich phase. These differ slightly from the compositions used in the simulation (Supplementary Table 1), which were based on tielines from earlier NMR measurements[26]. Because of system size, the probability of phase separation in the simulated systems is negligible and they can also be treated as homogeneous. One mole percent of 16:0–16 doxyl PC, a probe molecule with an N–O radical spin label located near the bilayer midplane, was added to DPPC to obtain an EPR resonance. The relaxation rates $R_1$ and $R_2$ are enhanced when the $O_2$ population near the N–O radical increases. The EPR measurement therefore measure oxygen populations near the midplane.

The chloroform was removed by spinning the solution in a glass vial while blowing nitrogen gas into the tube until dry. Samples were then placed under high vacuum for an hour to ensure complete removal of chloroform. Nitrogen exposed samples were prepared in a glove box filled with pure nitrogen and hydrated using degassed $H_2O$. A volume of 3 microliters of samples was filled into glass capillaries with an inner diameter of 0.87 mm yielding a sample height of 5 mm to achieve a

homogenous microwave $\mathbf{B_1}$ field strength over the sample in the resonator of the EPR instrument. Samples exposed to oxygen or air were exposed to the gases for an hour after initially being prepared under nitrogen. Tubes were sealed with Critoseal (McCormick Scientific, St. Louis, MO) at both ends and transferred to a 4-mm-quartz tube, filled with nitrogen, oxygen, or air, respectively, and sealed with a cap secured by Parafilm. Before running the EPR experiments, samples were stored at 50 °C for an hour to ensure homogeneity.

EPR spectra were recorded on an EMXnano spectrometer with a temperature control unit (Bruker BioSpin Corp, Billerica, MA) at a temperature of 25 °C, a microwave frequency of 9.63 GHz, a 100 kHz magnetic field modulation at an amplitude of 1 Gauss (G), a spectral width of 150 G, a microwave power of 5 mW, and a sweep time of 30 s. The linewidth of the $M_I = 0$ resonance was recorded at a spectral width of 20 G, an enhanced resolution of 0.01 G per data point, a microwave power of 1.6 mW, and a sweep time of 30 s. Power-saturation experiments on the $M_I = 0$ resonance were conducted at power levels from 50–0.05 mW and a reduced sweep time of 4 s to reduce sample heating.

The linewidth of the $M_I = 0$ resonance is reported as peak-to-peak width of the first derivative spectrum. For homogeneously broadened resonances, it is proportional to the transverse relaxation rate $R_2$. The power-saturation data were analyzed by fitting the peak-to-peak amplitude, $I$, of the first derivative spectrum to the formula[22]

$$I = I_0 \left( \sqrt{P} / \left( 1 + (2^{2/3} - 1) P/P_{1/2} \right)^{3/2} \right), \qquad (8)$$

where $I_0$ is a fit parameter reporting signal intensity, $P$ is the applied microwave power, and $P_{1/2}$ is a fit parameter that is proportional to the product of longitudinal and transverse relaxation rates $R_1 R_2$ by the linewidth $M_I = 0$ resonance yields a parameter proportional to $R_1$. Equation 8 assumes that the saturation of the $M_I = 0$ resonance follows a homogeneous saturation limit, which was confirmed experimentally.

**Evaluation of oxygen population from simulation**. The quantity $n_{O2}(r)$ is the oxygen population near the probe, whose location is assumed to be at the methyl carbon atom of chain 2 of DPPC in simulations. The oxygen population is defined as the number of oxygen atoms within a certain distance $r$ from the probe. In practice, the number of oxygens is counted and averaged over all time frames and over all DPPC molecules in the simulation.

$n_{O2}(r)$ was obtained from the radial distribution function $g(r)$ between the selected methyl carbon atom and oxygen. Specifically, the oxygen population is proportional to the integral of the radial distribution function

$$n_{O2}(r) \sim \int_0^r g(r') r' 2 dr', \qquad (9)$$

and is therefore equivalent to the coordination number of oxygen for the selected methyl carbon atom. The $g(r)$ functions in Supplementary Fig. 5 show that there are indeed oxygens present within a distance of 4–5 Å from the selected methyl carbon. The value of 4.5 Å was used for the simulation estimate shown in Fig. 8.

**Reporting summary**. Further information on research design is available in the Nature Research Reporting Summary linked to this Article.

## Data availability

Data supporting the findings of this paper are available from the corresponding authors upon reasonable request. A Reporting Summary for this Article is available as a Supplementary Information file. The source data underlying Figs. 2–8, Table 1, Supplementary Figs. 1, 2, 5–8, and Supplementary Tables 4, 5, 7 provided as a Source Data file.

## Code availability

Code for the Bayesian analysis is available at https://github.com/annekegh/mcdiff.

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

## Acknowledgements

We thank Gerald Feigenson, Ilya Leventhal, and John Nagle for helpful comments, Oriana De Vos for technical assistance, and the reviewers for their careful readings and valuable suggestions. This research was supported in part by Ghent University (A.G.), the Intramural Research Program of the National Institutes of Health, National Heart, Lung and Blood Institute (R.M.V. and R.W.P.), the National Institute on Alcohol Abuse and Alcoholism (W.E.T and K.G.), and NIH R01 GM120351 (E.L.). The computational resources and services used in this work were provided by the VSC (Flemish Super-computer Center), funded by the Research Foundation—Flanders (FWO), the Flemish Government—department EWI, and the high-performance computational capabilities at the National Institutes of Health, Bethesda, MD (NHLBI LoBoS cluster and CIT Biowulf

clusters). Anton 2 computer time was provided by the Pittsburgh Supercomputing Center (PSC) through Grant R01GM116961 from the National Institutes of Health.

## Author contributions

A.G., R.W.P., and K.G. designed the research, A.G. and R.M.V. carried out the simulations, K.G. and W.E.T. carried out the experiments; all authors analyzed the data and wrote the paper.

## Competing interests

The authors declare no competing interests.
