## [Peer Review File · Nature Communications]

Reviewers' Comments:

Reviewer #1:

Remarks to the Author:

Summary:

The paper addresses pathways of oxygen and water diffusion in liquid-ordered and liquid-disordered lipid bilayers, in an attempt to provide insight into the structure and function of lipid rafts. The work is of high quality, and the evidence supports the most important claims made. Two findings are of particular interest: (1) oxygen and water penetrate liquid-ordered bilayer leaflets along boundaries between ordered and disordered phases; and (2) both oxygen and water molecules travel along the liquid-ordered bilayer midplane, facilitating full crossing of the bilayer as well as some access of these molecules to less ordered domains. The work is impactful, relative to understanding diffusive processes in biological membranes, which consist of multiple components (diverse lipids and proteins) and exhibit variable fluidity according to the local composition and structure. The authors emphasize potential relevance of the work for understanding putative lipid rafts. In addition, the work is certainly relevant to understanding nanoscale variations in membrane structure and function. Within the biophysics community, there is a good deal of current interest in understanding membrane heterogeneities, especially with regard to local ordering effects. Examining the influence of such heterogeneities on small molecule permeation is a valid and interesting line of inquiry. The claims of the paper are supported by several lines of evidence, including a key point of experimental validation. Error analysis is handled appropriately, and known force field errors are accounted for in the data interpretation. Overall, care in generating and interpreting the data is evident. The work is insightful, within the realm of oxygen and water transit across liquid-ordered and liquid-disordered bilayers, and the link to lipid rafts is generally well-supported, though proteins are notably omitted from the model membranes (a common practice that can, nonetheless, be insightful). This is a valuable, interesting, and timely study that I would like to see in the literature.

I am honored to have the opportunity to provide a review, and I offer my comments in the spirit of enhancing the value of the manuscript to the community.

Primary concerns and clarifications:

1. Close to the end of the Discussion and Conclusions section, the statement is made, "The structure and dynamics of the Lo phase elucidated here indicate that rafts are quite porous, and that an anesthetic would be able to rapidly diffuse into the more ordered DPPC microdomains along the midplane...."
 - o It seems like a big leap to make a direct claim about the porosity of rafts, based on a highly simplified model that lacks membrane proteins and many of the lipid constituents of biological rafts (such as sphingolipids).
 - o While DPPC is a key component of the model Lo phase, is it equally prominent as a component of biological rafts? If not, the reference to "DPPC microdomains" in the context of rafts may be misplaced.
 - o It would be appropriate to qualify the very broad reference to "an anesthetic," as general anesthetics come in a wide range of sizes and may or may not diffuse in the same way as oxygen and water (notably small permeants).
2. Equation 4 is missing in the copy of the manuscript I received, so I have not been able to evaluate fully the comparison between the EPR data and the MD simulations.
3. A large amount of data and supplemental methods are provided in the supplementary information (SI). This complicates the paper and seems to force the reader to consult the SI to understand the primary argument of the paper. In at least one place (second paragraph of page 13), a supplemental figure is referenced in enough detail that the reader cannot follow the thread of the argument without accessing the SI.
4. Several figures and tables are misreferenced in the text (using incorrect figure/table numbers), and there are numerous other typographical errors. A thorough proofreading is required. Some specific errors or possible errors:

- o First sentence of Results: 375 ps is stated as the time interval, but the figure caption states 500 ps.
 - o Page 9, second paragraph: "The Lo is thicker than the Lo."
 - o Figure 6 caption: "center peak in Fig. 6" - seems to refer to Figure 5.
 - o Page 17: "Table 3 shows the excellent agreement..." - seems to refer to Table 2.
 - o Page 18: "This is shown most clearly in the bottom panel of Fig. 2" - seems to refer to Figure 1.
 - o Page 20, bottom: "Those from the BA followed the same trends but were slightly lower *than*..."
 - o Page 21, top: $\tau_{esc} \approx \tau_{esc}$ - should be $\tau_{entr} \approx \tau_{esc}$?
 - o In SI: "The dividing surface is reported in Tab. 2 of the main document." - seems to refer to Table 4.
 - o SI page 6: "The values of K for [missing words] listed in Table..."
 - o Figure S5 caption: An "orange" line is referenced in the top left panel, but I think the line is black.
 - o SI page 15: Figure 7 is referenced, but I think the authors mean Figure 5.
5. In Table S4, The oxygen partition coefficient, K, values for both the Lo and Ld phase models (0.38 and 0.68) seem quite a bit lower than expected from the authors' recent simulation work, where a hexadecane/water partition coefficient of about 16 was calculated [A. Ghysels, R.M. Venable, R.W. Pastor, G. Hummer, Position-dependent diffusion tensors in anisotropic media from simulation: Oxygen transport in and through membranes, *J. Chem. Theory Comput.*, 13 (2017) 2962–2976.]. Is this an error, or am I missing something?
6. It would be helpful to the reader for the Introduction to discuss the model systems used to represent the Ld and Lo phases. Unless the reader proceeds to the Methods before continuing to the Results and Discussion, a misconception can develop that the Ld and Lo phases are observed in the same bilayer (as is suggested by the term "phase"). In fact, the Ld and Lo phases have been modeled independently, in separate bilayer systems. Knowing this is important to understanding the results.

Stylistic comments:

- The abstract and introduction begin by substantiating the relevance of studying liquid-ordered bilayers for understanding raft structure and function. The manuscript would be strengthened by raising a broader (biological) question regarding the significance of membrane-order heterogeneities for the permeation of small molecules. Alternatively, the broader significance of rafts might be discussed before establishing the validity of a liquid-ordered bilayer model for studying them. In particular, sentence 2 of the abstract seems like an odd starting point for the paper: "...the functional implications of this correspondence [between rafts and the liquid-ordered phase of three-component bilayers] have only begun to be explored." It seems the functional significance of the rafts, themselves, would be of primary interest, rather than their correspondence with the liquid-ordered phase.
- A lot of results are discussed in the Introduction, which seems unusual. I think the authors did this to provide background to the hypothetical models presented. Still, I find it unsettling that a good deal of the Introduction is devoted to results and discussion.
- The concluding sentence of the main text (Discussion and Conclusions) could be strengthened by recapping the key findings.

Other concerns and clarifications:

- Near the end of the abstract, the reference to electron paramagnetic resonance should indicate what technique was used and at what location/depth in the bilayer the observed ratio of oxygen concentrations was found.
- Page 3: References would be appropriate to substantiate the statement, "Recent experiments indicate that the essential structural features of rafts and Lo phases are shared by localized regions in the plasma membrane."
- Scant reference is made to prior literature regarding molecular dynamics study of membrane permeability. While I recognize that the authors recently published an extensive review in this area, a few more citations would be appropriate. In particular, Equation 1 is closely related to an

approach used earlier by de Groot, Hub, and colleagues [J.S. Hub, B.L. de Groot, Does CO₂ permeate through aquaporin-1?, *Biophys. J.*, 91 (2006) 842–848.], who should be cited.

- Results - Last sentence of first paragraph on page 6: The average time of H₂O exit is stated. I am curious what the time is for O₂.

In the following sentence, $F(z)$ is mentioned, but I think it is not defined previously in the text, as the Methods section appears after the Results.

- Figure on page 7 (and related figures): Are the purple O₂ molecules imaged from a single snapshot of the simulation system, along with the lipids?

- Page 9 (top): What are the "assumptions inherent in the BA" that may influence the permeability values, in comparison with the counting method?

- Page 9, third paragraph: " $D_{\text{parallel}}(z)$ for the Lo in the chain regions is systematically lower (as much as a factor of 6)" - may be misleading because the profiles do not line up along the bilayer depth axis. There is a systematic shift "outward" for the Lo bilayer because of its greater thickness.

- Page 13, beginning of first paragraph: "The fraction of 3-DPPC..." Is this sentence referring to the profile for O₂?

- Page 13, last paragraph: Do you mean the "single" nearest-neighbor analysis (Fig. S6), as opposed to the 3 nearest-neighbor analysis?

Please clarify the statement "... shows that water interacts relatively more strongly with cholesterol in the Lo phase than does oxygen." Does the evidence support a *stronger* interaction, or could a *more frequent* interaction also generate the same data? The word "strongly" implies knowledge of the interaction energy, which I am not sure is supported by the data.

- Page 15, second paragraph: The reader is directed to the SI, "(see SI)," but this reference is not clear. Where in the SI?

- Page 15, last paragraph: "The reason for switching [from N₂] to air [rather than pure oxygen] was to limit relaxation enhancement." Using the bracketed words would help clarify the meaning.

- Discussion and Conclusions - Page 22, second paragraph: "The simulations also provide insight into the probe location..." I am not sure I understand the argument here. Are you saying that because the free energy wells are equally deep, the probes are in the same place?

- Methods - Page 26: " r is the transition rate..." - Please clarify what the "transition rate" is. I think you have used the term "crossing rate" elsewhere.

- Page 27, bottom: "The preceding profiles..." This reference is unclear.

- Page 30: What is I_0 ?

- Page 31, top: It seems a reference to the SI is needed, to understand $n_{\text{O}_2}(r)$.

- SI page 5: A correction is discussed to account for early transitions of oxygen molecules into the bilayer. However, it is not stated how the correction was applied. What was done, exactly?

- SI page 11: The nearest-neighbor analysis presented here seems somewhat redundant, although it is explained nicely (maybe more clearly than the related explanation in the main text). A sentence at the beginning of the section would help to differentiate this analysis from the 3 nearest-neighbor analysis in the main text.

- SI page 13: r' is not defined.

Overall evaluation:

This is intriguing work, with a lot of insights into the permeation process, especially where ordered membrane domains are present. I recommend publication, following revision.

Signature of reviewer:

Sally C. Pias, New Mexico Institute of Mining and Technology

Reviewer #2:

Remarks to the Author:

This paper determined the permeability of oxygen and water in model bilayers mimicking lipid rafts

using molecular dynamics (MD) simulations. The Lo and Ld phases of ternary lipid bilayers, such as DPPC/DOPC/cholesterol, were characterized with different parameters using MD simulations, including permeability, free energy, and diffusion profiles. In addition, the MD data were compared with experimental oximetry using electron paramagnetic resonance (EPR). The authors describe a model that allows small permeants to translocate through the boundary regions of the lipid-raft leaflets and freely diffuse along the bilayer midplane before exiting the membrane. The study is addressing an important scientific question with a broad interest to the community.

Major points:

1). Page 31: the method of $I(r)$ calculation is missing, including Eq.4 that is referenced in the Results section.

2). The lipid ratios used in the MD simulations and EPR experiments are different. For MD, DPPC/DOPC/cholesterol 0.55/0.15/0.30 was used as the Lo phase (Table 5), however 0.58/0.09/0.33 was used for the EPR studies. Could the authors justify or clarify this?

3). Page 30 (main text) and Page 15 (SI): the authors assumed that $P_{1/2}$ is proportional to R_1 relaxation rate. However, $P_{1/2}$ is proportional to $R_1 \cdot R_2$. Since the authors compared quantitatively the R_1 rates of the samples with significantly different linewidths, i.e., Lo and Ld phases, spectral linewidth (R_2) needs to be taken into account for the R_1 rate comparison. The references include Oh et al., *Methods Mol. Bio.*, 145:147-69, 2000; Altenbach et al., *PNAS*, 91:1667-1671, 1994.

4). Different oxygen concentrations, such as 21% and 100%, were used for R_1 , R_2 , and $I(r)$ comparisons. Could the authors provide justifications?

Minor points:

5). Fig. 2: there are shoulders at 20 to 30 Å in the top panel. Is there an explanation for it?

6). Page 13, line 15, 16: could the authors locate the "with oxygen" data in the statement "The top row of Fig. S5 shows that the PMFs for water in DPPC/DOPC/chol are virtually identical with and without oxygen (left panel)"?

7). Page 18, line 21, 22: the authors need to cite the supporting data or references for the statement "In contrast, the behaviour of both oxygen and water in the liquid disordered phase is similar to that in homogeneous DOPC at the same temperature..."?

8). Page 22, line 15: " $\langle z \rangle = 3.0$ Å in Lo and 2.4 Å in Ld" needs a reference or supporting data.

9). Editing errors:

Page 3, line 2 from the bottom: Lo and Ld phases.

Page 5, line 9: synthesize

Page 9, line 9: it is

Page 9, line 11: is thicker than Ld

Page 13, line 6: a right parenthesis is missing.

Page 16: The molar ratio for the Lo is not correct.

Page 18, line 14: Fig. 1 (not Fig. 2)

Page 21, line 4: $\text{resc} \approx \text{tentr}$

Page 22, line 20: replace the dot with a comma

Page S6, line 4: delete "for"

Reviewer #3:

Remarks to the Author:

This manuscript describes in great detail simulations and experiments investigating the presence and permeation pathways of oxygen and (to a lesser extent) water in bilayer membranes containing either Lo or Ld phases. The most surprising, and important, result is that the Lo phase is fairly permeable to these compounds, unlike the gel phase which has fairly similar chain order and area per molecule. The detailed analysis of MD trajectories indicates the pathway of oxygen permeation, and this too is surprising – that O₂ apparently diffuses laterally in the bilayer midplane before exiting the bilayer. Simulations are bolstered by ESR measurements conducted either in the presence or absence of oxygen, demonstrating that O₂ is present in the midplane in both Lo and Ld membranes.

I find this to be a valuable contribution that is carefully executed and presented. My major concern is that I believe that the version of the manuscript that I read is missing a key section of the methods section that describes how the ESR measurement is converted to $I(r)$ (including an equation 4 that is mentioned in the main text). This is a key part of interpreting this data. Moreover, I think that this section should be included in results, at least in an abbreviated form, so that a reader can make sense of the later figures without having to dig into the detailed methods. Right now I do not know what to make of Fig 7.

I have a few smaller comments:

1. I find that the second half of the introduction (starting on the top of page 4) reads more like it belongs in the discussion section, as the results of the manuscript are used in the arguments made here. I think it would be better to use this real estate to better introduce why such a detailed study of O₂ and water permeability is justified. An argument is made at the end of the discussion section currently, but an abbreviated argument here would give the results higher impact.

2. It is my understanding that most simulations were conducted on either Lo or Ld mixtures of DOPC/DPPC/Chol, though this was expanded on starting at the middle of page 13, when the authors re-analyze trajectories simulated for past publications. I find the naming scheme that comes in at this point somewhat confusing, since generally associate three component mixtures with ones that contain coexisting phases, although I realize that is not what is done here. Maybe there is a less confusing way to discuss these different lipid compositions? Also, what is the main goal of including the analysis on other mixtures? I think this should be more explicitly stated in the results section. Is it to demonstrate that the results with DPPC likely translate to these other systems?

3. Fig 4: it is stated in the text that there is reduced statistics for water though this is not highlighted explicitly in the plots. I am guessing this is seen in the extra wiggles/reduced symmetry compared to O₂. Is there a way to make this more explicit in the plot? Include error bounds or confidence intervals?

4. Figs 5,6: it would be nice to include a legend within the plots in addition to explaining the points in the text.

5. are binding sites for permeating molecules in proteins preferentially in the bilayer mid-plane?

Response to Reviewer Comments for NCOMMS-19-25869.

We thank you for your careful readings, positive comments, and valuable suggestions. They greatly improved the manuscript.

Reviewer #1 (Remarks to the Author):

Summary:

The paper addresses pathways of oxygen and water diffusion in liquid-ordered and liquid-disordered lipid bilayers, in an attempt to provide insight into the structure and function of lipid rafts. The work is of high quality, and the evidence supports the most important claims made. Two findings are of particular interest: (1) oxygen and water penetrate liquid-ordered bilayer leaflets along boundaries between ordered and disordered phases; and (2) both oxygen and water molecules travel along the liquid-ordered bilayer midplane, facilitating full crossing of the bilayer as well as some access of these molecules to less ordered domains. The work is impactful, relative to understanding diffusive processes in biological membranes, which consist of multiple components (diverse lipids and proteins) and exhibit variable fluidity according to the local composition and structure. The authors emphasize potential relevance of the work for understanding putative lipid rafts. In addition, the work is certainly relevant to understanding nanoscale variations in membrane structure and function. Within the biophysics community, there is a good deal of current interest in understanding membrane heterogeneities, especially with regard to local ordering effects. Examining the influence of such heterogeneities on small molecule permeation is a valid and interesting line of inquiry. The claims of the paper are supported by several lines of evidence, including a key point of experimental validation. Error analysis is handled appropriately, and known force field errors are accounted for in the data interpretation. Overall, care in generating and interpreting the data is evident. The work is insightful, within the realm of oxygen and water transit across liquid-ordered and liquid-disordered bilayers, and the link to lipid rafts is generally well-supported, though proteins are notably omitted from the model membranes (a common practice that can, nonetheless, be insightful). This is a valuable, interesting, and timely study that I would like to see in the literature.

I am honored to have the opportunity to provide a review, and I offer my comments in the spirit of enhancing the value of the manuscript to the community.

Primary concerns and clarifications:

1. Close to the end of the Discussion and Conclusions section, the statement is made, "The structure and dynamics of the Lo phase elucidated here indicate that rafts are quite porous, and that an anesthetic would be able to rapidly diffuse into the more ordered DPPC microdomains along the midplane...."

- o It seems like a big leap to make a direct claim about the porosity of rafts, based on a highly simplified model that lacks membrane proteins and many of the lipid constituents of biological rafts (such as sphingolipids).

o While DPPC is a key component of the model L_o phase, is it equally prominent as a component of biological rafts? If not, the reference to "DPPC microdomains" in the context of rafts may be misplaced.

o It would be appropriate to qualify the very broad reference to "an anesthetic," as general anesthetics come in a wide range of sizes and may or may not diffuse in the same way as oxygen and water (notably small permeants).

Response (to preceding three points): Two sets of simulations (SI) did contain sphingomyelin and the results were essentially identical to the simulations with DPPC presented in the main text. That stated, we agree with the preceding three points, and have modified this section of text as follows:

The structure and dynamics of the L_o phase elucidated here suggest that rafts are quite porous, and that a small anesthetic would be able to rapidly diffuse into the more ordered DPPC or PSM microdomains along the midplane and more effectively disrupt the phase. It is clear, of course, that a wide range of other permeants needs to be examined to assess the generality of the observations presented here. Furthermore, proteins present in cellular rafts but not included in the present simulations would be expected to modulate the permeation pathways for certain anesthetics.

2. Equation 4 is missing in the copy of the manuscript I received, so I have not been able to evaluate fully the comparison between the EPR data and the MD simulations.

Response: We apologize for the missing equation. Following the suggestion of Reviewer 3 it has been moved to the Results and is now Eq. 1.

3. A large amount of data and supplemental methods are provided in the supplementary information (SI). This complicates the paper and seems to force the reader to consult the SI to understand the primary argument of the paper. In at least one place (second paragraph of page 13), a supplemental figure is referenced in enough detail that the reader cannot follow the thread of the argument without accessing the SI.

Response: We moved the figure noted above describing the pathway analysis for PSM containing systems to the main text (it is the new figure 5). We agree that it makes the main text easier to read.

In contrast, the also-large Fig S6 in the original submission and the explanation of the nearest-neighbor analysis does require a serious excursion to the SI. Given the reviewers comments we decided that the original Fig S6 does not add significantly to the essential arguments presented in the main text so have deleted this figure and accompanying text from the SI.

4. Several figures and tables are misreferenced in the text (using incorrect figure/table numbers),

and there are numerous other typographical errors. A thorough proofreading is required.

Some specific errors or possible errors:

o First sentence of Results: 375 ps is stated as the time interval, but the figure caption states 500 ps.

o Page 9, second paragraph: "The Lo is thicker than the Lo."

o Figure 6 caption: "center peak in Fig. 6" - seems to refer to Figure 5.

o Page 17: "Table 3 shows the excellent agreement..." - seems to refer to Table 2.

o Page 18: "This is shown most clearly in the bottom panel of Fig. 2" - seems to refer to Figure 1.

o Page 20, bottom: "Those from the BA followed the same trends but were slightly lower *than*..."

o Page 21, top: $\tau_{esc} \approx \tau_{esc}$ - should be $\tau_{entr} \approx \tau_{esc}$?

Response: All fixed. We are very grateful for the careful read.

o **In SI:** "The dividing surface is reported in Tab. 2 of the main document." - seems to refer to Table 4.

Response: fixed.

o SI page 6: "The values of K for [missing words] listed in Table..."

Response: This has been replaced with "The values of K listed in Table..."

o Figure S5 caption: An "orange" line is referenced in the top left panel, but I think the line is black.

Response: Correct. It's black.

o SI page 15: Figure 7 is referenced, but I think the authors mean Figure 5.

Response: fixed.

5. In Table S4, The oxygen partition coefficient, K, values for both the Lo and Ld phase models (0.38 and 0.68) seem quite a bit lower than expected from the authors' recent simulation work, where a hexadecane/water partition coefficient of about 16 was calculated [A. Ghysels, R.M. Venable, R.W. Pastor, G. Hummer, Position-dependent diffusion tensors in anisotropic media from simulation: Oxygen transport in and through membranes, *J. Chem. Theory Comput.*, 13 (2017) 2962–2976.]. Is this an error, or am I missing something?

Response: The referee missed nothing, and we are grateful for such a careful reading. We had missed a unit conversion, and our partition coefficients were underestimated by a factor of 10. The revised Table S4 is reproduced below, where it can be seen that K are much more in line with our previous results for hexadecane/water and, indeed, experiment.

The numbers in the main text were essentially unchanged (some rounding) because they were ratios. Furthermore, there was no such mistake in c_w so the permeabilities were not affected.

6. It would be helpful to the reader for the Introduction to discuss the model systems used to represent the L_d and L_o phases. Unless the reader proceeds to the Methods before continuing to the Results and Discussion, a misconception can develop that the L_d and L_o phases are observed in the same bilayer (as is suggested by the term "phase"). In fact, the L_d and L_o phases have been modeled independently, in separate bilayer systems. Knowing this is important to understanding the results.

Response: The Results section now begins with the following description of the systems, and emphasizes that they are not mixtures of the two phases:

As detailed further in the Methods, the simulated and experimental systems presented here are of homogeneous liquid ordered and liquid disordered phases of DPPC/DOPC/cholesterol; i.e., they are not mixtures of the two phases. However, while the lipids in the L_d phase are distributed

Permeant	c_w (nm ⁻³)		c_m (nm ⁻³)		K	
	L_o	L_d	L_o	L_d	L_o	L_d
O ₂	0.0191 (0.0033)	0.0096 (0.0005)	0.0669 (0.0008)	0.0624 (0.0001)	3.5 (0.7)	6.5 (0.4)
water	30.6 (0.0056)	28.0 (0.0020)	6.071 (0.006)	7.416 (0.003)	0.1980 (0.0002)	0.2651 (0.0001)

relatively uniformly, the DPPC in the L_o phase forms hexagonally packed microdomains.

Stylistic comments:

- The abstract and introduction begin by substantiating the relevance of studying liquid-ordered bilayers for understanding raft structure and function. The manuscript would be strengthened by raising a broader (biological) question regarding the significance of membrane-order heterogeneities for the permeation of small molecules. Alternatively, the broader significance of rafts might be discussed before establishing the validity of a liquid-ordered bilayer model for studying them. In particular, sentence 2 of the abstract seems like an odd starting point for the paper: "...the functional implications of this correspondence [between rafts and the liquid-ordered phase of three-component bilayers] have only begun to be explored." It seems the functional significance of the rafts, themselves, would be of primary interest, rather than their correspondence with the liquid-ordered phase.

Response: We have edited the first sentence of the Abstract as follows:

Increasing experimental evidence supports the existence of ordered nanodomains (or rafts) in cholesterol rich eukaryotic cell membranes. However, their functional significance has only begun to be explored. This study exploits the correspondence of cellular rafts and the liquid ordered (L_o) phase of three-component lipid bilayers to examine permeability of small molecules.

- A lot of results are discussed in the Introduction, which seems unusual. I think the authors did this to provide background to the hypothetical models presented. Still, I find it unsettling that a good deal of the Introduction is devoted to results and discussion.

Response: Much of the “previewing” of results has been removed from the Introduction and the entire presentation of the simple models has been moved to the beginning of the Discussion.

- The concluding sentence of the main text (Discussion and Conclusions) could be strengthened by recapping the key findings.

Response: I love that last sentence.

Other concerns and clarifications:

- Near the end of the abstract, the reference to electron paramagnetic resonance should indicate what technique was used and at what location/depth in the bilayer the observed ratio of oxygen concentrations was found.

Response: The sentence has been expanded as follows (length restrictions for the Abstract do not allow more detail):

Electron paramagnetic resonance using a 16:0-16 PC Doxyl spin probe indicates that the ratio of oxygen concentrations near the midplanes of the L_d and L_o phases

- Page 3: References would be appropriate to substantiate the statement, "Recent experiments indicate that the essential structural features of rafts and L_o phases are shared by localized regions in the plasma membrane."

Response: The studies supporting this claim are were provided in the remainder of the paragraph. To make this more clear, references (6-9) are included in the sentence noted as follows:

Recent experiments⁶⁻⁹ indicate ...

- Scant reference is made to prior literature regarding molecular dynamics study of membrane permeability. While I recognize that the authors recently published an extensive review in this area, a few more citations would be appropriate. In particular, Equation 1 is closely related to an approach used earlier by de Groot, Hub, and colleagues [J.S. Hub, B.L. de Groot, Does CO₂ permeate through aquaporin-1?, Biophys. J., 91 (2006) 842–848.], who should be cited.

Response: We have added this and two other citations to counting methods

Yang, L. W. & Kindt, J. T. Simulation Study of the Permeability of a Model Lipid Membrane at the Fluid Solid Phase Transition. *Langmuir* **31**, 2187-2195, (2015)

de Groot, B. L., Tieleman, D. P., Pohl, P. & Grubmüller, H. Water Permeation through Gramicidin A: Desformylation and the Double Helix: A Molecular Dynamics Study. *Biophys. J.* **82**, 2934-2942, (2002).

The suggestion to include more citations to membrane simulations of permeation has been answered in two ways:

1. The following passage in the Introduction where the notion that oxygen and water are good candidates for a simulation-based study of permeability:

Oxygen and water are excellent candidates for such a study. They are small and their permeation can thereby be simulated directly using conventional MD, as opposed to enhanced sampling methods¹⁰ that are not yet validated for liquid ordered phases. Their hydrophobicities span a wide range and include many permeants of general interest. Lastly, their permeabilities have been extensively simulated in fluid (L_α) phase lipid bilayers, informing the comparison with L_o and L_d phases. In particular, recent simulations have examined the modulation of oxygen permeability by different lipids¹¹⁻¹³ and cholesterol¹⁴⁻¹⁶, and have probed the effects of potential energy functions on water permeability^{17,18}.

2. The former first subsection of the Discussion has now been split. The first subsection, *Model of Permeation in Liquid Ordered Phases*, focuses on the qualitative aspects of the model. A new second subsection of the Discussion, *Permeation in Liquid Disordered Phases*, notes the similarity of the oxygen results with fluid phase simulations from other groups:

In contrast to the L_o phase, the behavior of both oxygen and water in the liquid disordered phase is similar to single component fluid phase bilayers at the same temperature. For example, the permeability of oxygen from the BA for the L_d (27.1 ± 1 cm/s) brackets the 27.5 cm/s recently calculated for a DOPC bilayer at the same temperature using the same methodology¹². The free energy and diffusion profiles are also similar to those obtained in recent simulations by other groups using different simulation programs and potential energy functions¹⁴⁻¹⁶, providing additional confidence in the comparison. Likewise, the permeability of water from counting for the L_d ($0.9 \pm 0.2 \times 10^{-3}$ cm/s) is only slightly lower than the $1.5 \pm 0.4 \times 10^{-3}$ cm/s for pure DOPC¹⁰. The 3 nearest-neighbor analysis (Table 4, bottom) indicates little preference for a given lipid type, consistent with the fact that the permeant does not need to avoid dense microdomains.

- Results - Last sentence of first paragraph on page 6: The average time of H₂O exit is stated. I am curious what the time is for O₂.

Response: As listed in Table 4, the exit time for O₂ is 62 ns. These times are considered in detail in the Discussion section. We have added the following pointer:

However, unlike oxygen, **which on average remains in the midplane for approximately 60 ns**, water is unstable in the midplane and, on average, escapes the bilayer in 700 ps. **A detailed analysis of the oxygen and water characteristic times is presented in the Discussion.**

In the following sentence, $F(z)$ is mentioned, but I think it is not defined previously in the text, as the Methods section appears after the Results.

Response: The definition of $F(z)$ has been moved from the Methods to the Results.

- Figure on page 7 (and related figures): Are the purple O₂ molecules imaged from a single snapshot of the simulation system, along with the lipids?

Response: Yes. This is now explicitly stated in the caption as follows:

Middle panel: side-view with DOPC and cholesterol removed, and with other oxygen molecules **from the initial configuration** (purple);

- Page 9 (top): What are the "assumptions inherent in the BA" that may influence the permeability values, in comparison with the counting method?

Response: the following new text has been added:

Differences in the individual values of P between the two methods may arise from assumptions inherent in the BA, which assumes one-dimensional diffusive transport without solvent memory effects. In contrast, the MD trajectories explicitly contain non-diffusive behavior at short time scales and memory effects associated with slow solvent relaxation.¹¹

- Page 9, third paragraph: " $D_{\text{parallel}}(z)$ for the Lo in the chain regions is systematically lower (as much as a factor of 6)" - may be misleading because the profiles do not line up along the bilayer depth axis. There is a systematic shift "outward" for the Lo bilayer because of its greater thickness.

Response: We agree that the quantitative comparison can be confusing (and distracting) and have removed the "factor of six" parenthetical comment.

- Page 13, beginning of first paragraph: "The fraction of 3-DPPC..." Is this sentence referring to the profile for O₂?

Response: It's for both O₂ and water, as implied by the preceding sentence. The reference to the barrier for O₂ has been deleted to prevent confusion:

The profiles for oxygen and water in the L_o phase are remarkably similar given the increased statistical error in the water counts. The fraction of 3-DPPC is approximately 0.45 at $|z| = 25 \text{ \AA}$ (the approximate location of phosphate ~~peak and the barrier for O₂~~) indicating no strong lipid preference **at the bilayer surface**.

- Page 13, last paragraph: Do you mean the "single" nearest-neighbor analysis (Fig. S6), as opposed to the 3 nearest-neighbor analysis?

Response: We did mean "single nearest-neighbor" analysis. However, in the spirit of reducing the SI, this figure and the accompanying text have been deleted

Please clarify the statement "... shows that water interacts relatively more strongly with cholesterol in the L_o phase than does oxygen." Does the evidence support a *stronger* interaction, or could a *more frequent* interaction also generate the same data? The word "strongly" implies knowledge of the interaction energy, which I am not sure is supported by the data.

Response: We appreciate this distinction. In the case of the hydrogen bond of water to the hydroxyl in the headgroup it's quite clearly an interaction energy. It's not obvious for the chain region and we agree that "more frequent" would be more appropriate. However, because this figure is now deleted (as noted above), the present passage has been deleted as well.

- Page 15, second paragraph: The reader is directed to the SI, "(see SI)," but this reference is not clear. Where in the SI?

Response: It is now stated explicitly: (see **Section 2 of the** SI for more details).

- Page 15, last paragraph: "The reason for switching [from N₂] to air [rather than pure oxygen] was to limit relaxation enhancement." Using the bracketed words would help clarify the meaning.

Response: The sentence was rewritten as follows:

The reason for **measuring in air (rather than pure oxygen)** was to limit relaxation enhancement.

- Discussion and Conclusions - Page 22, second paragraph: "The simulations also provide insight into the probe location..." I am not sure I understand the argument here. Are you saying that because the free energy wells are equally deep, the probes are in the same place?

Response: We have clarified this section. The free energy wells in Fig. 2 give information about the oxygen concentrations only, not about the probe location. We now explicitly state that the

average probe location is calculated from the MD trajectories. It turns out that the average probe location is not equal to $z=0$. Rather, the probe samples the oxygen concentration in the region 2 to 4 Angstrom (where the oxygen concentrations differ substantially according to Fig. 2). The new text is as follows:

The simulations also provide insight into the probe location, assuming that the terminal methyl group of the chain is an adequate surrogate of the probe. Specifically, the free energy well for oxygen is almost equally deep at $z = 0$ (Fig. 2, top), meaning that the local concentration is approximately the same at the very center of the L_d and L_o phases. Nevertheless, the ratio is statistically different from 1 in both experiment and simulations. This originates from the flexibility of the probe location, and that it is not located exactly at the center of the membrane. Instead, the average positions of the C16 atom of the C2 chains in DPPC calculated directly from the MD trajectories are displaced from the midplane, with $\langle z \rangle = 3.0 \text{ \AA}$ in L_o and $\langle z \rangle = 2.4 \text{ \AA}$ in L_d , indicating that they sample different parts of the free energy profiles. Furthermore, the oxygen free energy profiles differ substantially 2-4 \AA from the midplane. Hence, it is not surprising that the concentration ratios are statistically different from 1 in simulations.

- Methods - Page 26: "r is the transition rate..." - Please clarify what the "transition rate" is. I think you have used the term "crossing rate" elsewhere.

Response: The details regarding r were specified in the SI to conserve space. This is made more clear in the revised passage:

... where r is the crossing rate of the permeant (see Eq. S2), ...

- Page 27, bottom: "The preceding profiles..." This reference is unclear.

Response: The passage has been clarified as follows:

The diffusion and free energy profiles yield assorted characteristic relaxation times and distances²³ ...

- Page 30: What is I_0 ?

Response: It is defined in the new text (which describes a different method suggested by R2):

The power saturation data were analyzed by fitting the peak-to-peak amplitude, I , of the first derivative spectrum to the formula²²

$$I = I_0 \left(\sqrt{P} / (1 + (2^{2/3} - 1)P/P_{1/2})^{3/2} \right) \quad (5)$$

where I_0 is a fit parameter reporting signal intensity, P is the applied microwave power, and $P_{1/2}$ is a fit parameter that is proportional to the product of longitudinal and

transverse relaxation rates $R_1 R_2$. Division of $P_{1/2}$ by the linewidth $M_I = 0$ resonance yields a parameter proportional to R_1 . Eq. (5) assumes that the saturation of the $M_I = 0$ resonance follows a homogeneous saturation limit, which was confirmed experimentally.

- Page 31, top: It seems a reference to the SI is needed, to understand $n_{O_2}(r)$.

Response: The following pointer is provided at the end of the relevant paragraph:

See Section 1.7 of the SI for more details on the explicit evaluation of $n_{O_2}(r)$ from the simulation.

- SI page 5: A correction is discussed to account for early transitions of oxygen molecules into the bilayer. However, it is not stated how the correction was applied. What was done, exactly?

Response: The modification that is discussed on SI page 5 had been applied in the presented results. To make this more clear, we have rewritten the paragraph as follows.

The evaluation of oxygen crossings requires a modification of the above procedure. Indeed, most of the permeants are already in the membrane after equilibration and consequently, the first exit of such an oxygen is not counted as a transition. This leads to an underestimate of the rate from Eq. (2). Therefore, a practical solution to this problem is applied for oxygen molecules as follows. The permeants that are inside the bilayer at the start of the simulation, have entered the bilayer from one particular side. It is now assumed that the oxygen molecules loose memory inside the membrane and randomly exit on either the same side or the other side. Hence, the initial escape of such a permeant out of the membrane will increase the total number of crossings by 0.5, on average. The counting of the oxygen crossings was modified to let those initial escapes count for only a 0.5 crossing instead of 1 crossing. This modification is not necessary for water because the water population inside the membrane at the start of each simulation is negligible.

- SI page 11: The nearest-neighbor analysis presented here seems somewhat redundant, although it is explained nicely (maybe more clearly than the related explanation in the main text). A sentence at the beginning of the section would help to differentiate this analysis from the 3 nearest-neighbor analysis in the main text.

Response: In the interest of reducing the need to reference the SI, the nearest neighbor analysis (the former Section 1.5 and Fig S6 has been deleted.

- SI page 13: r' is not defined.

Response: r' is the integration variable and therefore does not need a definition. The appearance of $r'^2 dr'$ in the integral is due to the volumetric integration over space, $\sin(\theta) r'^2$

$d(\phi) d(\theta) dr'$. This simplifies to $r^2 dr'$ because of spherical symmetry.

Overall evaluation:

This is intriguing work, with a lot of insights into the permeation process, especially where ordered membrane domains are present. I recommend publication, following revision.

Reviewer #2 (Remarks to the Author):

This paper determined the permeability of oxygen and water in model bilayers mimicking lipid rafts using molecular dynamics (MD) simulations. The L_o and L_d phases of ternary lipid bilayers, such as DPPC/DOPC/cholesterol, were characterized with different parameters using MD simulations, including permeability, free energy, and diffusion profiles. In addition, the MD data were compared with experimental oximetry using electron paramagnetic resonance (EPR). The authors describe a model that allows small permeants to translocate through the boundary regions of the lipid-raft leaflets and freely diffuse along the bilayer midplane before exiting the membrane. The study is addressing an important scientific question with a broad interest to the community.

Major points:

1). Page 31: the method of $I(r)$ calculation is missing, including Eq.4 that is referenced in the Results section.

Response: We apologize for the missing equation. Following the suggestion of Reviewer 3 it has been moved to the Results and is now Eq. 1, and the details are now much easier to follow.

2). The lipid ratios used in the MD simulations and EPR experiments are different. For MD, DPPC/DOPC/cholesterol 0.55/0.15/0.30 was used as the L_o phase (Table 5), however 0.58/0.09/0.33 was used for the EPR studies. Could the authors justify or clarify this?

Response: The following justification has been added after the experimental composition are listed:

The preceding compositions were based on recent neutron scattering results (Dorell et al., *in preparation*) and assures that each phase is homogenous; i.e., there is no L_d in the DPPC rich phase and no L_o in the DOPC rich phase. These differ slightly from the compositions used in the simulation (Table 5) which were based on tielines from earlier NMR measurements²³. Because of system size, the probability of phase separation in the simulated systems is negligible so these can also be treated as homogeneous.

3). Page 30 (main text) and Page 15 (SI): the authors assumed that $P_{1/2}$ is proportional to R_1 relaxation rate. However, $P_{1/2}$ is proportional to $R_1 \cdot R_2$. Since the authors compared quantitatively the R_1 rates of the samples with significantly different linewidths, i.e., L_o and L_d phases, spectral linewidth (R_2) needs to be taken into account for the R_1 rate comparison. The

references include Oh et al., *Methods Mol. Bio.*, 145:147-69, 2000; Altenbach et al., *PNAS*, 91:1667-1671, 1994.

Response: We are grateful for this suggestion, and have reanalyzed all of our EPR data using the more precise method described in the references above. While values of R1 and R2 changed slightly, the ratios in the Ld and Lo phase were nearly identical (old = 1.6 ± 0.5 ; new = 1.74 ± 0.35). The new ratios are reported in the text. The new method is introduced in the Results (page 11), detailed in the Methods (page 25), and the raw data is presented in the SI (page 13); all of the new text is in red. (It seemed unwieldy to paste in it this response.)

4). Different oxygen concentrations, such as 21% and 100%, were used for R1, R2, and I(r) comparisons. Could the authors provide justifications?

Response: The sentence present in the original manuscript was expanded as follows:

The reason for **measuring in air (rather than pure oxygen)** was to limit relaxation enhancement.

Minor points:

5). Fig. 2: there are shoulders at 20 to 30 Å in the top panel. Is there an explanation for it?

Response: This is the effect of the ordered DPPC chains. It is not as prominent for water because the free energy is already so high. This is now noted in the text as follows:

the shoulders between |10-20| Å reflect the ordering of the acyl chains.

6). Page 13, line 15, 16: could the authors locate the “with oxygen” data in the statement “The top row of Fig. S5 shows that the PMFs for water in DPPC/DOPC/chol are virtually identical with and without oxygen (left panel)”?

Response: The entire paragraph has been rewritten as follows:

The top row shows that $F_w(z)$ for L_o and L_d phases with *N-palmitoyl sphingomyelin* and either DOPC/chol or POPC/chol (top row, middle and right panels, respectively) are very similar to those for DPPC/DOPC/chol. The left panel includes results from the simulations carried out with oxygen (also shown in Fig. 2). The $F_w(z)$ for each phase are virtually identical, indicating that oxygen is not perturbing the structures. The 3-neighbor analyses (middle and lower rows of Fig. S5) show qualitatively similar correspondences between DPPC and PSM bilayers, though the statistical errors are higher because of the lower number of saved coordinate sets. In particular, the ordered subdomains of DPPC and PSM exclude water in their chain regions but allow in the midplane.

7). Page 18, line 21, 22: the authors need to cite the supporting data or references for the statement “In contrast, the behaviour of both oxygen and water in the liquid disordered phase is similar to that in homogeneous DOPC at the same temperature...”?

Response: This particular point has been given its own subsection as follows:

Permeation in Liquid Disordered Phases. In contrast to the L_o phase, the behavior of both oxygen and water in the liquid disordered phase is similar to fluid phase bilayers at the same temperature. For example, the permeability of oxygen from the BA for the L_d (27.1 ± 1 cm/s) brackets the 27.5 cm/s recently calculated for a similarly DOPC bilayer at the same temperature and using the same methodology¹². The free energy and diffusion profiles are also similar to those obtained in recent simulations by other groups using different programs and potential energy functions¹⁴⁻¹⁶, providing additional confidence in the comparison. Likewise, the permeability of water from counting for the L_d ($0.9 \pm 0.2 \times 10^{-3}$ cm/s) is only slightly lower than the $1.5 \pm 0.4 \times 10^{-3}$ cm/s for pure DOPC¹⁰. The 3 nearest-neighbor analysis (Table 4, bottom) indicates little preference for a given lipid type, consistent with the fact that the permeant does not need to avoid dense microdomains.

Additionally, the following newly named subsection, **Comparison of Liquid Ordered, Liquid Disordered, and Fluid Phases**, presents a systematic comparison of the three phases, with numerical data in Tables 3 and 4 (these were present in the original ms, but not as well integrated).

8). Page 22, line 15: “ $\langle z \rangle = 3.0$ Å in L_o and 2.4 Å in L_d ” needs a reference or supporting data.

Response: These values were calculated directly from the trajectory. This is now stated explicitly as:

... the average positions of the C16 atom of the C2 chains in DPPC calculated directly from the trajectories are slightly further from the midplane in L_o ($\langle z \rangle = 3.0$ Å) than in the L_d ($\langle z \rangle = 2.4$ Å) implying that the probe samples different parts of the free energy profiles.

9). Editing errors:

Page 3, line 2 from the bottom: L_o and L_d phases.

Page 5, line 9: synthesize

Page 9, line 9: it is

Page 9, line 11: is thicker than L_d

Page 13, line 6: a right parenthesis is missing.

Page 16: The molar ratio for the L_o is not correct.

Page 18, line 14: Fig. 1 (not Fig. 2)

Page 21, line 4: $\tau_{esc} \approx \tau_{entr}$

Page 22, line 20: replace the dot with a comma

Response: All fixed. We are very grateful for the careful read.

Page S6, line 4: delete “for”

Response: fixed.

Reviewer #3 (Remarks to the Author):

This manuscript describes in great detail simulations and experiments investigating the presence and permeation pathways of oxygen and (to a lesser extent) water in bilayer membranes containing either Lo or Ld phases. The most surprising, and important, result is that the Lo phase is fairly permeable to these compounds, unlike the gel phase which has fairly similar chain order and area per molecule. The detailed analysis of MD trajectories indicates the pathway of oxygen permeation, and this too is surprising – that O₂ apparently diffuses laterally in the bilayer midplane before exiting the bilayer. Simulations are bolstered by ESR measurements conducted either in the presence or absence of oxygen, demonstrating that O₂ is present in the midplane in both Lo and Ld membranes.

I find this to be a valuable contribution that is carefully executed and presented. My major concern is that I believe that the version of the manuscript that I read is missing a key section of the methods section that describes how the ESR measurement is converted to I(r) (including an equation 4 that is mentioned in the main text). This is a key part of interpreting this data. Moreover, I think that this section should be included in results, at least in an abbreviated form, so that a reader can make sense of the later figures without having to dig into the detailed methods. Right now I do not know what to make of Fig 7.

Response: Following the reviewer’s suggestion, we have moved the section related to estimation of the EPR from simulation from the Methods to the Results (pp 18-19). We attempted an abbreviated form, but it was still confusing and repetitive. The more technical details of some of the estimates remain in the SI.

I have a few smaller comments:

1. I find that the second half of the introduction (starting on the top of page 4) reads more like it belongs in the discussion section, as the results of the manuscript are used in the arguments made here. I think it would be better to use this real estate to better introduce why such a detailed study of O₂ and water permeability is justified. An argument is made at the end of the discussion section currently, but an abbreviated argument here would give the results higher impact.

Response: The “previewing” of results has been removed from the Introduction and the presentation of the simple models has been moved to the beginning of the Discussion. This

newly vacated land allowed the inclusion of the following passage explaining why oxygen and water are used here:

Oxygen and water are excellent candidates for such a study. They are small and their permeation can thereby be simulated directly using conventional MD, as opposed to enhanced sampling methods¹⁰ that are not yet validated for liquid ordered phases. Their *hydrophobicities span a wide range and include many permeants of general interest. Lastly, their permeabilities have been extensively simulated in fluid (L_α) phase lipid bilayers, informing the comparison with L_o and L_d phases. In particular, recent simulations have examined the modulation of oxygen permeability by different lipids¹¹⁻¹³ and cholesterol¹⁴⁻¹⁶, and have probed the effects of potential energy functions on water permeability^{17,18}.*

2. It is my understanding that most simulations were conducted on either L_o or L_d mixtures of DOPC/DPPC/Chol, though this was expanded on starting at the middle of page 13, when the authors re-analyze trajectories simulated for past publications. I find the naming scheme that comes in at this point somewhat confusing, since generally associate three component mixtures with ones that contain coexisting phases, although I realize that is not what is done here. Maybe there is a less confusing way to discuss these different lipid compositions? Also, what is the main goal of including the analysis on other mixtures? I think this should be more explicitly stated in the results section. Is it to demonstrate that the results with DPPC likely translate to these other systems?

Response: The Results section now begins with the following description of the systems, and emphasizes that they are not mixtures of the two phases:

As detailed further in the Methods, the simulated and experimental systems presented here are of homogeneous liquid ordered and liquid disordered phases of DPPC/DOPC/cholesterol; i.e., they are not mixtures of the two phases. However, while the lipids in the L_d phase are distributed relatively uniformly, the DPPC in the L_o phase forms hexagonally packed microdomains.

The section on PSM-containing liquid ordered phases has been rewritten to make it easier to follow and the figure describing the 3 nearest-neighbor analysis has been moved from the SI to the main text. This analysis shows more clearly that the results for DPPC do translate to these other systems. The revised text is as follows:

The water analysis was also performed on previously published²² simulations carried out on the Anton supercomputer²⁷ with no added oxygen (see Table S1 for compositions and other system details). In four of the six of these systems, PSM takes the role of the DPPC; the substitution of POPC for DOPC is also investigated. The relative lipid concentrations in these ternary systems yield homogeneous L_o or L_d phases. The free energy profiles are very similar to the two DPPC/DOPC/chol systems already described (Fig. 5 top). In particular, $F_w(z)$ for DPPC/DOPC/chol with and without oxygens are virtually identical indicating that oxygen is not perturbing the structures. The 3-neighbor analyses (Fig. 5 middle, bottom) also shows the same behavior: the ordered subdomains of DPPC and PSM exclude water in their chain regions but allow it in the midplane.

3. Fig 4: it is stated in the text that there is reduced statistics for water though this is not highlighted explicitly in the plots. I am guessing this is seen in the extra wiggles/reduced symmetry compared to O₂. Is there a way to make this more explicit in the plot? Include error bounds or confidence intervals?

Response: The plots are quite crowded at this point. As the reviewer notes, the reduced symmetry is a good measure of the uncertainty. We note this in the following sentence:

Results of the 3 nearest-neighbor analysis are plotted in Fig. 4. The profiles for oxygen and water in the L_o phase are remarkably similar given the increased statistical error in the water counts (this is evident in the asymmetry of the profiles).

4. Figs 5,6: it would be nice to include a legend within the plots in addition to explaining the points in the text.

Response: Done.

5. are binding sites for permeating molecules in proteins preferentially in the bilayer mid-plane?

Response: Not in general. Ion channels typically have a series of binding sites distributed along the channel. The selectively filter of water and gas channels can be anywhere. However, the more relevant protein for oxygen permeation in membranes is cytochrome C oxidase. The oxygen binding site (the end of a tunnel from the membrane to the protein center) is located directly at the bilayer midplane, as consistent with the high population of oxygen in this region. However, this example not relevant to permeation in the liquid ordered phase, and we have already noted this in our previous papers in oxygen permeation in fluid phases. Hence, we have not modified the manuscript.

Reviewers' Comments:

Reviewer #1:

Remarks to the Author:

The authors have addressed all of my concerns, and I recommend publication of the manuscript. Two minor corrections are suggested below, relating to text introduced during the revision process.

1. Page 23 of the main text states, "These differ slightly from the compositions used in the simulation (Table 5) which were based on tielines from earlier NMR measurements." I am unfamiliar with the word "tielines," in this context. Did the authors mean "timelines"?

2. In the SI, page 5: "loose memory" should read "lose memory."

Sally C. Pias

Associate Professor of Chemistry

New Mexico Institute of Mining and Technology

Reviewer #2:

Remarks to the Author:

The manuscript has been improved from the original submission. The authors addressed the comments raised by the reviewers. Most of the concerns of this reviewer have been resolved. From my perspective, the paper can be considered for publication after correcting the following minor errors.

1). Page 11, line 9: "The additional broadening of the [missing word] for"

2). Page 13, line 2: "See Section 1.7 of the SI"; should it be section 1.6?

3). SI page 2, title: "SIMULATONS" is misspelled.

4). SI, page 3, line 3 from the bottom: "(3) and [missing value] at 310K as reported previously."

Reviewer #3:

Remarks to the Author:

My comments have been largely addressed.

I still find the final figure to be confusing. Perhaps it would be useful to include the relevant points from Table 2 in the figure itself so that the reader can easily see the agreement stated in the text? It would also be good to make clear in the first sentence of the caption that the analysis presented is from simulations (vs. experiments). I gather that the 5A length for comparison comes from the quenching length-scale referenced in results. If so, this would be good to state explicitly either in the caption or in the results text that describes this figure (or both).

Response to Reviewer Comments for NCOMMS-19-25869B.

We thank you again for your careful readings, positive comments, and valuable suggestions. They continue improve the manuscript.

Reviewer #1 (Remarks to the Author):

The authors have addressed all of my concerns, and I recommend publication of the manuscript. Two minor corrections are suggested below, relating to text introduced during the revision process.

1. Page 23 of the main text states, "These differ slightly from the compositions used in the simulation (Table 5) which were based on tielines from earlier NMR measurements." I am unfamiliar with the word "tielines," in this context. Did the authors mean "timelines"?

Response: We did mean tielines. This is a term associated with 3-component phase diagrams.

2. In the SI, page 5: "loose memory" should read "lose memory."

Response: corrected, thanks.

Reviewer #2 (Remarks to the Author):

The manuscript has been improved from the original submission. The authors addressed the comments raised by the reviewers. Most of the concerns of this reviewer have been resolved. From my perspective, the paper can be considered for publication after correcting the following minor errors.

- 1). Page 11, line 9: "The additional broadening of the [missing word] for"
- 2). Page 13, line 2: "See Section 1.7 of the SI"; should it be section 1.6?
- 3). SI page 2, title: "SIMULATONS" is misspelled.
- 4). SI, page 3, line 3 from the bottom: "(3) and [missing value] at 310K as reported previously."

Response: corrected, thanks.

Reviewer #3 (Remarks to the Author):

My comments have been largely addressed.

I still find the final figure to be confusing. Perhaps it would be useful to include the relevant

points from Table 2 in the figure itself so that the reader can easily see the agreement stated in the text? It would also be good to make clear in the first sentence of the caption that the analysis presented is from simulations (vs. experiments). I gather that the 5Å length for comparison comes from the quenching length-scale referenced in results. If so, this would be good to state explicitly either in the caption or in the results text that describes this figure (or both).

Response: This is a great idea. The new figure and caption (below) take all of these points into account.

Fig. 8 Relative oxygen population near bilayer midplanes. Left panel: Simulated $I(r)$, the ratio of (normalized) oxygen population near the probe in the L_d and L_o phases (equation **Error! Reference source not found.**) as a function of the interaction radius r between the methyl carbon of chain 2 of DPPC and oxygen, with $I(r = 4.5 \text{ \AA})$ shown explicitly (filled square). The blue band is twice the standard error over the 4 replicas. Right panel: Experimental ratios of $R_{1,Ld}/R_{1,Lo}$, $R_{2,Ld}/R_{2,Lo}$, and their average (open circles). The uncertainties are standard errors from the fits. The Source data are provided as a Source Data file.